# SIRA: Exposing Vulnerabilities in Text Watermarking with Self-Information Rewrite Attacks

## Abstract

Text watermarking is designed to embed hidden, imperceptible, markers within content generated by large language models (LLMs), with the goal of tracing and verifying the content's origin to prevent misuse. The robustness of watermarking algorithms has become a key factor in evaluating their effectiveness, but remains an open problem. In this work, we introduce a novel watermark removal attack, the Self-Information Rewrite Attack (SIRA), which poses a new challenge to the robustness of existing watermarking techniques. Since embedding watermarks requires both concealment and semantic coherence, current methods prefered to embed them in high-entropy tokens. However, this reveals an inherent vulnerability, allowing us to exploit this feature to identify potential green tokens. Our approach leverages the self-information of each token to filter potential pattern tokens that embed watermarks and performs the attack through masking and rewriting in a black-box setting. We demonstrate the effectiveness of our attack by implementing it against seven recent watermarking algorithms. The experimental results show that our lightweight algorithm achieves state-of-the-art attack success rate while maintaining shorter execution times and lower computational resource consumption compared to existing methods. This attack points to an important vulnerability of existing watermarking techniques and paves way towards future watermarking improvements.

## 1 Introduction

Large language models (LLMs), exemplified by ChatGPT (OpenAI, 2024) and Claude (Anthropic, 2024), have demonstrated remarkable capabilities in generating coherent, human-like text. However, while these advances significantly expand AI's potential in content creation, they have concurrently heightened concerns regarding their misuse (Deshpande et al., 2023; Wang et al., 2024), including the spread of misinformation (Monteith et al., 2024) and threats to academic integrity (Stokel-Walker, 2022).

To mitigate risks associated with LLM-generated content, text watermarking has emerged as a promising countermeasure (Kirchenbauer et al., 2023; Aaronson & Kirchner, 2022). This technique subtly alters the LLM's generation process to embed imperceptible patterns in the output text, which, while invisible to human readers, can be reliably detected using specialized algorithms. This generate-detect framework enables differentiation between AI-generated and human-authored content and allows tracing the text back to the specific LLM that created it (Li et al., 2024). Consequently, this mechanism promotes accountability and helps mitigate LLM misuse, providing a reliable means to ensure transparency and integrity in AI-generated content.

Recent studies have demonstrated that watermarking techniques exhibit significant robustness against simple manipulations, including word deletions (Welbl et al., 2020) and emoji attacks (Kirchenbauer et al., 2023). However, traditional NLP attack strategies, such as token deletion and insertion, are increasingly insufficient for thoroughly evaluating the robustness of advanced watermarking algorithms. As LLMs continue to advance, there is a growing need for more sophisticated testing methodologies that account for complex manipulation tactics, ensuring that watermarking techniques remain resilient against emerging threats. To provide a more rigorous evaluation of

watermarking robustness, model-based attacks have been proposed. These methods typically rewrite watermarked texts using either closed-source commercial LLMs (Kirchenbauer et al., 2023) or fine-tuned open-source models trained on specially designed phrase datasets (Krishna et al., 2024). Despite their potential, these approaches face several limitations. First, they lack transparency, offering limited control over the output text via a small set of hyperparameters. Second, they are often costly, requiring token-based payments or significant computational resources. Furthermore, for newer and more robust watermarking algorithms like SIR (Liu et al., 2024), these methods fail to deliver effective attacks.

To resolve the aforementioned problem, we propose a lightweight and effective watermark removal attack named **SIRA**(self-information rewrite attack). This method is inspired by a simple intuition: watermarking algorithms aim to be imperceptible to users while preserving text quality, often favoring high-entropy tokens to embed watermark patterns (Kirchenbauer et al., 2023; Liu et al., 2023). The high-entropy tokens, however, will usually have high self-information. Leveraging this insight, we can identify potential "green list" token candidates within watermarked text under a black-box setting. By masking these potential tokens and allowing the LLM to complete the masked segments, we can effectively carry out a more effective paraphrasing attack. SIRA requires minimal resources and works effectively even with a small model like LLaMA3-8B. Our experiments in seven watermark algorithms show that SIRA outperforms all other black-box watermark removal attacks. Specifically, our method achieves over a 90% attack success rate on most current watermarking techniques (Kirchenbauer et al., 2023; Zhao et al., 2023; Liu et al., 2023; Lu et al., 2024; Wu et al., 2024; Aaronson & Kirchner, 2022). We will release our code to further promote the development of responsible AI (Gu, 2024).

## 2 RELATED WORK

**Self-information.** Self-information, also known as surprisal, is a fundamental concept in Information Theory, first introduced by Claude Shannon in his seminal work (Shannon, 1948). Shannon employed self-information as the principal metric to quantify the information content associated with the occurrence of specific events, effectively linking the rarity of an event to the amount of information it communicates. This measure forms a cornerstone in the understanding of data encoding and compression, as articulated in Shannon's source coding theorem, where events that are more probable are represented with fewer bits, while less probable events require more bits, thereby facilitating optimal compression (Shannon, 1948).

In the realm of Natural Language Processing (NLP), self-information plays a crucial role in the analysis and modeling of language. It aids in deciphering language patterns, particularly in evaluating the entropy and predictability of tokens within sequences. The concept is particularly useful for quantifying the informativeness or surprise of a token, determined by its probability in a given linguistic context. Language models, for instance, predict the probability of a subsequent token in a sequence using the preceeding context:

$$P(t_k \mid t_1, t_2, \ldots, t_{k-1})$$

The self-information of the token in this context is computed as follows:

$$I(t_k \mid t_1, t_2, \ldots, t_{k-1}) = -\log_b(P(t_k \mid t_1, t_2, \ldots, t_{k-1}))$$

where $I(t_k \mid t_1, t_2, \ldots, t_{k-1})$ denotes the self-information of token $t_k$ given the context of previous tokens, $P(t_k \mid t_1, t_2, \ldots, t_{k-1})$ is the probability of token $t_k$ occurring after the preceding sequence of tokens, and $b$ represents the base of the logarithm, typically set to 2. Tokens that are less predictable within a sequence, hence more informative, exhibit higher self-information values.

**LLM Watermark.** Watermarking techniques for large language models are designed to embed identifiable patterns in model outputs, allowing for the traceability of generated text back to its originating source. These watermarks serve as an essential tool for ensuring accountability and ownership, particularly in scenarios where identifying the specific model or version that produced the content is crucial. LLM watermark methods can be broadly classified into two primary categories: the KGW Family and the Christ Family. Each family employs distinct mechanisms that are

integral to the internal workings of LLMs. The *KGW Family* (Kirchenbauer et al., 2023; Liu et al., 2023; Zhao et al., 2023; Wu et al., 2024; Lu et al., 2024) focuses on modifying the logits—the raw output probabilities produced by the model—before they are transformed into text. This approach involves selectively adding bias to certain tokens, referred to as "green list" tokens, which influences the model to favor these tokens, thus embedding a statistical signature in the output. Post text generation, a statistical metric based on the proportion of these "green" tokens is computed. A predetermined threshold enables differentiation between watermarked and non-watermarked text.

Conversely, the *Christ Family* (Aaronson & Kirchner, 2022; Christ et al., 2024; Kuditipudi et al., 2023) modifies the sampling process during the generation phase itself. Rather than altering the logits, this family intervenes directly in the token selection during decoding. Techniques such as top-k sampling, temperature adjustment, or nucleus sampling are adapted to ensure preferential selection of watermarked tokens. This method provides more direct control over the generation process, embedding watermarks that are resilient against post-processing attacks, such as paraphrasing.

**Watermark Removal Attacks.** The robustness of a watermarking algorithm is crucial, as it determines the effectiveness of the watermark under various real-world conditions, particularly in adversarial settings. Attacks against watermarking algorithms, commonly referred to as watermark tampering attacks, can be broadly categorized into two types:

*Text manipulations*: These attacks involve traditional NLP techniques to straightforward text manipulations, such as word deletion (Welbl et al., 2020), substitution (Yu et al., 2010), or insertion (Kirchenbauer et al., 2023). By altering the surface-level structure of the text, these methods attempt to distort or eliminate the watermark without drastically changing the content's meaning. These techniques exploit the fact that many watermarking algorithms embed patterns with a pre-designed "green token" and "red token", making them vulnerable to such basic modifications. Kirchenbauer et al. (2023) propose emoji attack and copy-paste attack which insert emoji/human writtern text in the generated text to avoid detection. These methods are considered variants of text manipulations, however, they are easily thwarted by detectors equipped with content filters and often alter the semantics of the generated text which makes them inappropriate for real-world use.

*Model-based paraphrasing*: A more advanced form of attack involves using another LLM to paraphrase the watermarked content. This approach takes advantage of the LLM's ability to generate diverse rephrasings while maintaining the core meaning of the text. Krishna et al. (2024) propose DIPPER, a paraphrase generation model developed by fine-tuning T5-XXL (Raffel et al., 2020b) on an aligned paragraph dataset. This model has been widely adopted in recent watermark research (Zhao et al., 2023; Liu et al., 2024; Kuditipudi et al., 2023) to evaluate the robustness of watermarking algorithms. Zhang et al. (2023) propose the random walk attack, which utilizes Llama2-7B (Touvron et al., 2023) as the generation model, with T5-XXL (Raffel et al., 2020b) serving as a perturbation oracle to iteratively modify the watermarked text. In each iteration, RoBERTa-v3 large (Liu, 2019) is employed as a quality oracle to provide rewards, while GPT-3.5 OpenAI (2023) performs a final quality check on the generated text. However, since this method does not ensure that the paraphrased text retains the same semantics as the original watermarked text, it diverges significantly from the goals of typical paraphrase attacks.

Our proposed SIRA falls under the category of model-based paraphrasing attacks. However, unlike conventional paraphrasing techniques that generate entirely new sentences or passages based on instructions, SIRA selectively replaces potential green words in the watermarked text to provide a "neutral" template for rewriting. This targeted replacement approach enables more precise control over the consistency and semantics of the watermark-removed text. Additionally, SIRA can be effectively implemented using a lightweight model, enhancing its efficiency for practical applications.

## 3 METHODS

In this section, we detail SIRA attack formulation and implementation. First, we lay out the problem setting in Section 3.1, then we develop the details of the method in Section 3.2.

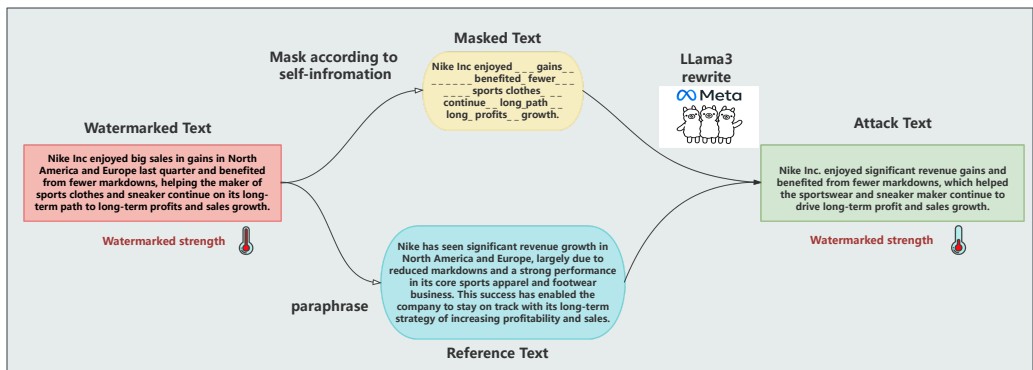

Figure 1: **SIRA pipeline consisting to two steps.** First, the attack generates a masked text based on self-information. If the self-information of a specific part above a pre-set threshold, that portion of the text is masked and replaced with a placeholder. Simultaneously, a reference text is generated by asking the LLM to paraphrase. In the second step, the LLM is prompted to complete the masked text while incorporating all the information from the reference text.

### 3.1 PROBLEM SETTING

**Definition 1 (Language generative model).** A Language generative model $M : X \rightarrow Y$ that maps any input prompt $x \in X$ to an output $y \in Y$, where $X$ the prompt space, $Y$ the output space. We denote $Y_h$ is human written text space, $Y_u$ is the machine generated unwatermarked text, $Y_w$ is the machine generated watermarked text.

**Definition 2 (Watermark Algorithm):** A watermark algorithm consists of a watermarking function $W$, a secret key $k$, and a detector $D$. The watermarking function $W$, parameterized by the key $k$, denote as $W_k$, modifies the output $y$ to embed a watermark, given an input prompt $x \in X$ resulting in a watermarked output $M(x, W_k) \rightarrow y_w \in Y_w$. The detector $D$, using the same key $k$, can then verify whether a given output $\hat{y} \in Y$ contains the embedded watermark. The detector $D$ operates as a binary classifier with the following output behavior:

$$D(W_k, \hat{y}) = \begin{cases} 1 & \text{if } \hat{y} \text{ is detected as watermarked} \\ 0 & \text{otherwise} \end{cases} \tag{1}$$

The detector $D$ contains a parameter $\theta$, where the $\theta$ is the z-score threshold.

**Definition 3 (Perturbation Function):** The attacker has a perturbation function $P : Y_w \rightarrow Y_p$ modifies the watermarked output $y_w$ to produce a perturbed output $y_p = P(y_w)$. The function $P$ aims to minimize the detection success rate of the detector $D$ on the perturbed output $y_p$. A function $S(y_w, y_p)$ measures the semantic similarity between the original watermarked output $y_w$ and the perturbed output $y_p = P(y_w)$. The pre-set threshold $\epsilon \in [0, 1]$ is a parameter that quantifies the minimum required level of semantic similarity between the original watermarked output $y_w$ and the perturbed output $y_p = P(y_w)$.

We define the scenario as a **black box adversarial problem** and we assume that the attacker **should not know the watermark algorithm $W$, the secret key $k$ and should not have access to the detector $D$.** The attacker does not have access to any information about the feature distribution of the watermark algorithm or the model architecture.

For watermark algorithm, the goal is to achieve a balance between robustness and performance. The detector $D$ is formulated as an optimization problem with the objective of minimizing classification errors. Specifically, the detector aims to maximize its accuracy in distinguishing between human-written text $y_h$ and attack text $y_p$. The goal of detector $D$ can represent as:

$$\max_{\theta_D} \quad \mathbb{E}_{y_h \sim Y_h} \left[ \log \left( 1 - D_{\theta_D}(W_k, y_h) \right] + \mathbb{E}_{y_p \sim Y_p} \left[ \log(D_{\theta_D}(W_k, y_p)) \right] \tag{2}$$

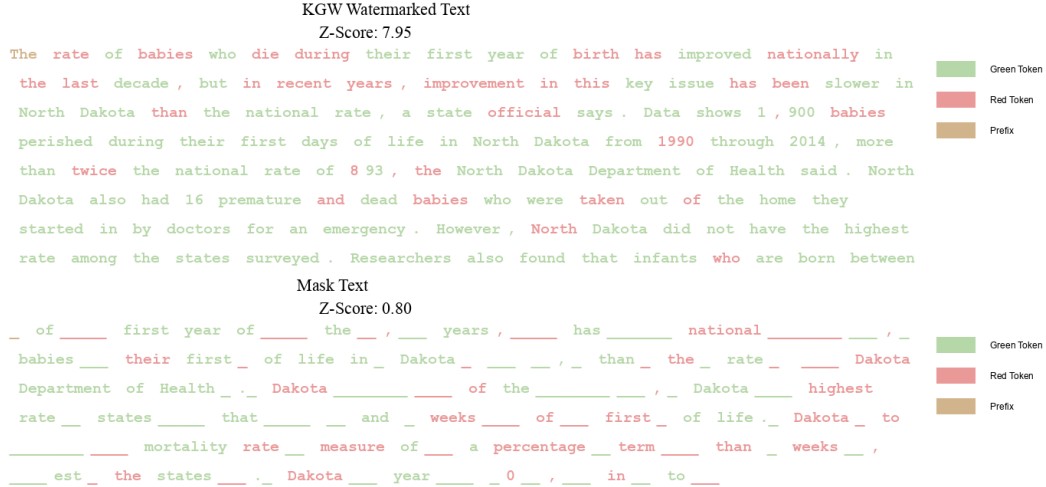

Figure 2: **Visualization of watermark text and the text after masking.** We can find that according to self-information filter, the mask text significantly reduce the z-score and remove most of the green tokens. Note that actual z-score is lower since some placeholder which will be removed in next step are treated as green token by detector. The shown text are using KGW-1 watermark.

For attacker, the perturbation function $P$ is defined to minimize the probability that the detector $D$ successfully identifies the watermark in the perturbed output $y_p$, while ensuring semantic preservation. The goal for $P$ can represente as:

$$P^* = \arg\min_P \quad \mathbb{E}\left[D(W_k, P(y_w))\right] \tag{3}$$

$$\text{s.t.} \quad S(y_w, P(y_w)) \geq \epsilon \tag{4}$$

Note that $D(W_k, P(y_w))$ is only used during the evaluation phase. The attacker does not have access to the detector during the training or generation stages.

## 3.2 SELF-INFORMATION REWRITE ATTACK

A primary challenge in watermark removal attacks is identifying the "green token" defined by the watermarking algorithm. Some methods, such as Random Walk (Zhang et al., 2023), use grammatical group matching to explicitly replace verbs. In contrast, approaches like DIPPER (Krishna et al., 2024) and GPT Paraphraser (Liu et al., 2024) delegate the task of rewriting and removing green token to large language models through high-level instructions. However, methods of this type lack transparency and control; relying on LLM for consistency with original watermarked text.

Our attack is based on a common principle of watermarking algorithms, as discussed in the KGW (Kirchenbauer et al., 2023; Liu et al., 2023) work: since the watermark must remain imperceptible to the user, high-entropy tokens are ideal candidates for embedding. High-entropy tokens exhibit a more uniform distribution of probabilities, this uniformity means that when logits are adjusted to increase the likelihood of green tokens, it is easier to embed watermarks effectively without significantly compromising the quality of the output. Meanwhile this also implied high-entropy token has lower probability thus higher self-information.

In our approach, we propose a straightforward and easily implementable solution by leveraging self-information to identify potential green-list tokens and subsequently rewrite them. High-entropy tokens are typically associated with high self-information due to their unpredictability and low probability of occurrence. Meanwhile, small probability changes caused by the watermark algorithm can reduce self-information, as briefly explained in Appendix F. By considering both the change in self-information and high-entropy token inherent nature, we classify tokens with high or moderate self-information as potential green-list tokens and filter them out to obtain a more neutral template for LLM rewriting. Empirically, our preliminary experiments show that utilizing self-information, rather than directly filtering based on high entropy, results in higher attack success rates.

---

**Algorithm 1** Pseudocode for Self-information rewrite attack

---

1: **Input:** Watermarked token sequence $\mathbf{y} = \{y_1, y_2, \ldots, y_n\}$, language model $M_{attack}$, self-information percentile $\epsilon$, instruction $\mathbf{s}$
2: **Output:** Response token sequence $\mathbf{y_p}$ without watermark.
3: $\mathbf{y}' \leftarrow M_{attack}(\mathbf{y})$          ▷ Paraphrase sequence $\mathbf{y}'$ using $M_{attack}$
4: $\mathbf{I} \leftarrow [\,]$
5: **for** $i = 1$ to $n$ **do**          ▷ Compute self-information for each token in $\mathbf{y}$
6:      $\mathbf{I}[i] \leftarrow -\log P(y_i \mid \text{context})$
7: **end for**
8: $\tau_\epsilon \leftarrow \text{Percentile}(\mathbf{I}, \epsilon)$          ▷ Determine threshold from $\epsilon$ percentile of $\mathbf{I}$
9: **for** $i = 1$ to $n$ **do**
10:      **if** $\mathbf{I}[i] > \tau_\epsilon$ **then**
11:          $y_i \leftarrow \varnothing$          ▷ Mask token if above threshold
12:      **end if**
13: **end for**
14: $\mathbf{y}_p \leftarrow M_{attack}(\mathbf{y}', \mathbf{y}, \mathbf{s})$          ▷ Generate de-watermarked response $\mathbf{y_p}$ using $M$
15: **return** $\mathbf{y_p}$

---

Given a watermarked text $y = \{y_0, y_1, \ldots, y_n\}$, where $y_i$ represents each token, we employ a base language model $M_{attack}$; $M_{attack}$ is distinct from the generative model $M$ used to produce the watermarked text. We use $M_{attack}$ to calculate the self-information for each token $y_t$ as follows:

$$I(y_t) = -\log P(y_t|y_0, y_1, \ldots, y_{t-1}; M_{attack}),$$

where $P(y_t|y_0, y_1, \ldots, y_{t-1}; M_{attack})$ denotes the probability of token $y_t$ given its preceding tokens in the sequence, as estimated by the language model $M_{attack}$. To mask the potential green list tokens, we set a threshold $\epsilon$, and get the overall paragraph threshold by percentile:

$$\tau_\epsilon \leftarrow \text{Percentile}(\mathbf{I}, \epsilon)$$

Any token with a self-information value $\mathbf{I}[i] > \tau_\epsilon$ is considered to be a potential token and will be masked and replaced with a placeholder. In our experiments, we discovered that using placeholders outperformed directly masking specific tokens. The placeholders serve as cues, maintain the text's structure, indicating where tokens have been masked whihc providing the LLM with hints about the original text's length and the likely number of words, allowing for more high quality reconstructions. We shown one visualization of mask text and watermark text comparsion in Figure 5.

However, the compression will still result in the loss of watermark text information details. To address this, we use the base LLM to rewrite the watermarked text, creating a reference text. This rewritten text serves as a reference to provide the semantic information intergrity during the second step. The reason we do not use the original watermarked text is that we find this leads LLM to take shortcuts: LLM tend to directly take the content from the watermark text, due to the high similarity between masked and watermark text.

In the final attack phase, we provide the LLM with the masked text, reference text, and instructions for a fill-in-the-blank task, guiding it to reconstruct the missing content. We provide the instructions we use in Appendix D. The pseduocode of our algorithm is shown in Algorithm 1.

## 4 EXPERIMENTS

### 4.1 SETUP

**Dataset and Prompts.** Following prior watermarking research (Kirchenbauer et al., 2023; Zhao et al., 2023; Liu et al., 2024; Kuditipudi et al., 2023), we utilize the C4 dataset (Raffel et al., 2020a) for general-purpose text generation scenarios. We selected 500 random samples from the test set to serve as prompts for generating the subsequent 230 tokens, using the original C4 texts as non-watermarked examples.

Table 1: Comparison of watermark algorithms under different attack methods.

| Comparison of Watermark Algorithms under Different Attack Methods | | | | | | | |
|---|---|---|---|---|---|---|---|
| Watermark ⟍ Attack | KGW-1 | Unigram | UPV | EWD | DIP | SIR | EXP |
| Word delete (Welbl et al., 2020) | 22.4% | 1.6% | 6.6% | 22.8% | 57.4% | 44.0% | 9.4% |
| Synonym Substitution (Yu et al., 2010) | 83.2% | 17.4% | 65.2% | 76.2% | 99.6% | 82.0% | 51.0% |
| GPT Paraphraser (Liu et al., 2024) | **100%** | 63.9% | 71.9% | 90.8% | **99.8%** | 58.8% | 72.2% |
| DIPPER-1 (Krishna et al., 2024) | 82.4% | 37.0% | 58.6% | 82.2% | 99.6% | 61.2% | 73.6% |
| DIPPER-2 (Krishna et al., 2024) | 95.8% | 45.6% | 61.8% | 89.0% | **99.8%** | 63.6% | 82.2% |
| SIRA(Ours) | **100%** | **93.8%** | **93.0%** | **100%** | **99.8%** | **83.4%** | **93.4%** |

**Watermark generation algorithms and language model.** To conduct a comprehensive evaluation, we select seven recent watermarking works: KGW (Kirchenbauer et al., 2023), Unigram (Zhao et al., 2023), UPV (Liu et al., 2023), EWD (Lu et al., 2024), DIP (Wu et al., 2024), SIR (Liu et al., 2024),EXP (Aaronson & Kirchner, 2022) in the assessment. The watermark hyperparameter settings shown in Appendix A, and the detection settings adhere to the default/recommendations (Pan et al., 2024) configurations of the original works. Specifically, for KGW-k, $k$ is the number of preceding tokens to hash. A smaller $k$ implies stronger attack robustness yet simpler watermarking rules. We use KGW-1 in our experiment. For language models, we follow previous work setting select (Kirchenbauer et al., 2023; Liu et al., 2024; Zhao et al., 2023) Opt-1.3B (Zhang et al., 2022) as watermark text generation model. We use LLaMA3-8b (Dubey et al., 2023) to ensure the lightweight and usability of our method, along with its tokenizer for the calculation of self-information.

**Baseline Methods.** For our method, we use $\epsilon = 0.3$ as threshold. For attack method, we use word deleteion (Welbl et al., 2020), synonym substitution (Yu et al., 2010), Dipper (Krishna et al., 2024), and GPT Paraphaser (Liu et al., 2024) to compare with our method. For GPT Paraphaser, we use the GPT-4o-2024-05-13 (OpenAI, 2024) version. For DIPPER-1 the lex diversity is 60 without order diversity, and for DIPPER-2 we additionally increase the order diversity by 40. The word deletion ratio is set to 0.3 and the synonym substitution ratio is set to 0.5. The synonyms are obtained from the WordNet synset (Miller, 1995).

**Evaluation.** We utilize the attack success rate as our primary metric. The attack success rate is defined as the proportion of generated attack texts for which the watermark detector incorrectly classify the attack text as unwatermarked sample, compared to the total number of attack texts. To mitigate the influence of detection thresholds, we follow prior work (Liu et al., 2024; Zhao et al., 2023) adjust z-threshold of detector until reach target false positive rate in Figure 3 . We use generated 500 attack texts as positive samples and 500 humnn-written texts as negative samples. We dynamically adjust the detector's thresholds to establish false positive rates at 1% and 10%, and we report the true positive rates and F1-scores. All experiments for our method were conducted on single NVIDIA A100 40GB GPU.

## 4.2 EXPERIMENTAL RESULTS.

In Table 1, we present the attack success rates of various watermark removal methods across different watermarking algorithms. The results demonstrate that our approach consistently outperforms all other methods for each watermarking algorithm. Notably, the closest competitors to our method are DIPPER (Krishna et al., 2024) and GPT Paraphraser (Liu et al., 2024), which are model-based paraphrasing attacks. Our approach surpasses these competitors by a significant margin in experiments involving seven watermarking algorithms on the C4 dataset (Raffel et al., 2020a).

To further demonstrate the effectiveness of our method and avoid the impact of a fixed z-threshold on detector performance, we follow previous work by setting the FPR to 1% and 10%, and report the true positive rate of the detector on adversarial texts based on the adjusted z-threshold corresponding to the FPR. Additionally, we report the best F1 score that the watermark algorithm can achieve

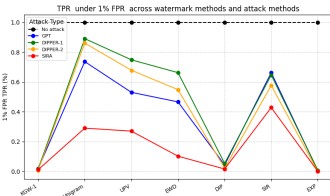 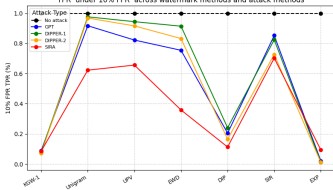 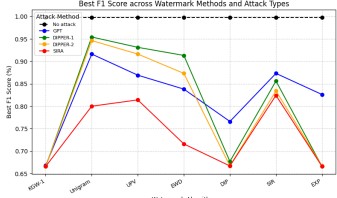

(a) True positive rate of different watermark with false postivite rate set to 1%.

(b) True positive rate of different watermark with false postivite rate set to 10%.

(c) Best F1-score could achieved by different watermark algorithm detector.

Figure 3: To avoid the default z-threshold impact the robustness of watermark algorithm, we dynamically adjust the z-score threshold of the watermarking algorithm until achieving specified false positive rates for the watermark detector. We display the true positive rate and the best F1 score could achieve by watermark detector. Lower TPR and F1 scores at a given FPR indicate that the watermark detector struggles more to differentiate between attack texts and human-written texts, suggesting a more effective attack. We show the detail number of above figure in Appendix B.

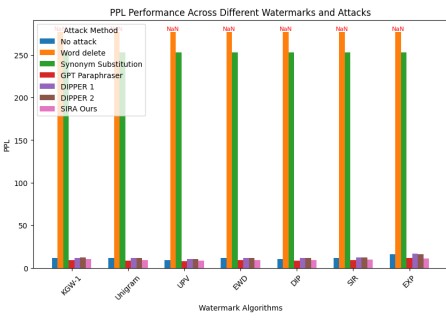 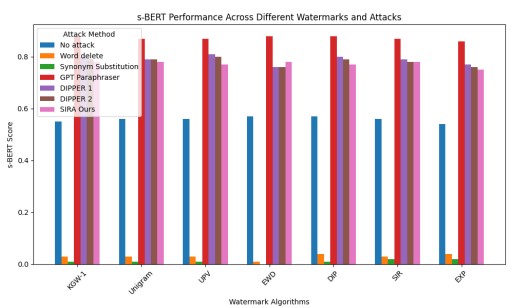

(a) Performance of PPL.

(b) Performance of s-BERT.

Figure 4: Performance comparison of watermark methods against various attack methods based on PPL (Perplexity↓) and s-BERT (Sentence-BERT score↑). The word delete will signicantly increase the PPL and lead to overflow. We marked the overflow data with NaN in the fig. 4a. The synonym substitution will also increase the PPL. The paraphrased text has better text quaility than original watermark text for our method and GPT Paraphraser.

under different attacks. The results are shown in Figure 3, and the detailed numbers are provided in Appendix B. Lower true positive at a given false positive rate indicate that the watermark detector struggles more to differentiate between adversarial texts and human-written texts. Our algorithm achieves optimal attack performance in most cases; suggesting a more effective attack.

## 4.3 TEXT QUALITY ANALYSIS

To further demonstrate that our method does not adversely affect text quality, we conduct additional evaluations of the text generated by the model. We compare **Perplexity (PPL)** of the text quality. Furthermore, we specifically calculate the sentence-level embedding similarity (**Sentence-BERT Score (s-BERT)**) before and after the attack to explore whether the attack alters the semantic content at the sentence level. We also conducted experiments in the Appendix E using ChatGPT as a judge to measure overall semantic similarity. The results, shown in Figure 4, indicate that our method has a smaller impact on text quality compared to other approaches.Our approach, similar to other model-based methods, benefits from more powerful large language models, achieving better performance in terms of the PPL metric compared to the original watermarked text. Additionally, our method retains a greater degree of semantic information. We show the detail numbers of two metrics in Appendix C.

Table 2: Comparison of Execution Time and VRAM Usage for Different Methods. Note that the execution speed of GPT Paraphraser may vary depending on the network status and real-time OpenAI server load.

| Method | Execution Time (s) | VRAM Usage (MB) |
|---|---|---|
| GPT Paraphraser | 12.8 | N/A |
| DIPPER | 14.7 | 45462 |
| SIRA | 10.3 | 18640 |

Table 3: Comparison with Random Masking Strategy. Notice here the random mask performance also benefit from other steps like rewriting in our framework. The vanilla random mask has similar attack success rate as word deletion.

| Mask Ratio | 0.4 | 0.5 | 0.6 | 0.7 | 0.8 |
|---|---|---|---|---|---|
| Random Mask | 52% | 66% | 78% | 80% | 82% |
| Self-information Mask | 80% | 88% | 92% | 96% | 100% |

## 4.4 GENERATION SPEED

In this section, We conducted the attack experiments using 50 distinct watermark texts, each containing approximately 230 ± 20 tokens. For each method, we measured both execution time and VRAM usage. The reported execution time reflects the average for a single attack instance. The experiments were run on two NVIDIA A100 GPUs, utilizing a sequential device map for baseline methods requiring multiple GPUs. The configuration for the GPT Paraphraser follows the setup described in Section 4.1. The results are shown in Table 2.

One of the main limitations of current model-based watermark removal attacks is their substantial resource consumption. For instance, DIPPER built on the T5-XXL model, necessitates two 40GB A100 GPUs for effective operation. Similarly, the GPT parser introduces considerable costs due to its dependence on a proprietary model that employs token-based billing.

In contrast, our proposed pipeline operates using a minimal configuration of the LLaMA3-8b model, requiring only 18GB of VRAM. This enables compatibility with many consumer-grade GPUs, significantly reducing hardware requirements. Our approach is efficient enough to run on GPUs with as little as 20GB of memory, and each attack is completed in just two model inferences which makes our attack faster than other methods. We also experimented with larger models in Appendix E, such as Llama3-80b, and found that using a larger model can further improve semantic preservation and the quality of generated text .

## 4.5 ABLATION EXPERIMENT

In this section, we aim to further scrutinize the self-information rewrite attack and emphasize the potential of this attack. We utilize Opt-1.3b and a random sample of 50 prompts from the C4 dataset to generate watermarked responses. Unless otherwise specified, we use Llama-3-8b as the base model for our attack. The temperature for the base model is set to 0.7.

**Self-information mask versus Random mask** In this experiment, we replace self-information-based selective masking with a random masking strategy, while keeping all other steps unchanged. We use the same masking ratios, ranging from 0.4 to 0.8 in increments of 0.1, and compare the resulting attack success rates. The Unigram watermarking method is employed to generate the watermarked text. The results are presented in Table 3. To ensure fair comparisons, the random masking strategy is executed five times, and the final average attack success rate is reported.

The results indicate that, at any given mask ratio, the self-information-based masking method significantly outperforms the random strategy. More importantly, the random masking approach exists a **bottleneck**, with limited improvement in attack success rates beyond a ratio of 0.6. This due to random mask can not make sure all target green token are removed. For a single watermarked text with fixed mask ratio, our method is deterministic, as the same tokens are masked each time. In contrast, the random approach does not provide this guarantee.

Table 4: Effect of self-information Threshold on the Success Rate of the UPV Algorithm

| self-information threshold $\epsilon$ | 0.25 | 0.30 | 0.35 | 0.40 | 0.45 | 0.5 | 0.55 | 0.60 | 0.65 | 0.70 |
|---|---|---|---|---|---|---|---|---|---|---|
| Attack Success Rate | 96% | 94% | 94% | 88% | 80% | 76% | 72% | 70% | 58% | 32% |

Table 5: Comparison of Attack Success Rate and Average z-score. The reference text is generated by asking the base model to paraphrase the watermarked response, while the attack text is generated using our two-step approach.

| Text | Attack Success Rate | Average z-score |
|---|---|---|
| Human-written Text | N/A | 0.12 |
| Reference Text | 64% | 3.75 |
| Attack Text | 94% | 1.85 |

**How does the self-information threshold affect final performance?** In this experiment, we use UPV as the watermarking algorithm. We varied the value of $\epsilon$ from 0.25 to 0.70 in increments of 0.05 to test its impact on the success rate of the attack using the UPV algorithm. The results are shown in table 4.

We observed that the attack success rate is directly influenced by the value of $\epsilon$. For the UPV algorithm, setting the threshold to 0.3 results in a highly effective attack. A significant performance gap is observed when $\epsilon$ increases from 0.60 to 0.65. Additionally, based on human evaluation, when $\epsilon$ is below 0.25, the generated attack text tends to lose more detailed information from the original watermarked text, such as character dialogues. Considering both performance and semantic preservation, we recommend setting $\epsilon$ between 0.2 and 0.3. For less robust algorithms, setting $\epsilon$ between 0.4 and 0.5 is sufficient to achieve an attack success rate exceeding 90%.

We set $\epsilon$ to 0.3 in Table 1 since watermarking algorithms may employ a hybrid strategy, embedding the watermark primarily in low-entropy tokens while embedding a smaller portion in high-entropy tokens or with a "hard list". Setting $\epsilon$ to 0.3 effectively removes the watermark while preserving the original semantics.

**Does the success of the attack due to paraphrased reference text?** We used the Unigram watermarking algorithm to generate watermarked text. We set the detector's z threshold to 4 according to its default settings. For a given input, the detector calculates its z-score, and if the score exceeds 4, the text is classified as watermarked.

We measured the attack success rate for each of the following stages: the reference text generated in the first step of our algorithm, and the final attack text. Additionally, we calculated the average z-score for each stage and reported the z-score of human-written text as a reference. The result are shown in Table 5. We observed that the attack success rate for the reference text is lower than that of the final attack text. Paraphrase strategies tend to preserve more n-grams from the original text, which may still be detectable by the watermark detection algorithm. In contrast, our attack reduces the presence of such n-grams by utilizing self-information filtering. Additionally, the z-score produced by our method is closer to that of human-written text compared to simple paraphrasing approaches.

## 5 CONCLUSION

In this paper, we presents the Self-Information Rewrite Attack (SIRA), a lightweight and effective method for removing watermarks from LLM-generated text by targeting anomalous tokens. Empirical results show that SIRA outperforms existing methods in attack success rates across multiple watermarking techniques while preserving text quality and requiring minimal computational resources. By exploiting vulnerabilities in current watermarking algorithms, SIRA highlights the need for more robust and adaptive watermarking approaches in watermark embedding. We will release our code to the community to facilitate further research in developing responsible AI practices and advancing the robustness of watermarking algorithms.

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

## BROADER IMPACT

In this work, we aim to provide an approach to test the robustness of Large Language Models watermark. We propose a method that can remove different watermark in LLM generated text. We are aware of the potential risks that our work entails for the security and safety of LLMs, as they are increasingly adopted in various domains and applications. Nevertheless, we also believe that our work advances the open and transparent research on the challenges and limitations of LLM watermark, which is crucial for devising effective solutions and protections. Similarly, the last few years the exploration of adversarial attacks (Wei et al., 2023; Madry et al., 2017; Krishna et al., 2024) has led to the improvement of responible AI and led to techniques to safeguard against such vulnerabilities,e further coordinated with them before publicly releasing our results. We also emphasize that, our ultimate goal in this paper is to identify of weaknesses of existing methods.

## LIMITATIONS

**Evaluation**:We have observed that existing watermarking techniques tend to embed in high-entropy text, which may represent a potential vulnerability. This characteristic could be exploited by attackers to weaken the security guarantees of LLMs. The work most similar to ours is DIPPER, but our motivations differ slightly: while DIPPER aims to evade detectors of AI-generated text, our focus is specifically on LLM watermarking algorithms. Due to this difference in objectives, we did not include methods for detecting AI-generated text, such as GPT-zero, in our evaluation. We will explore in future work whether our method can be extended to detectors for adversarial AI-generated text.

**Empirical**: One limitation of our work is that it is specifically tailored to natural language. Some watermarking methods for LLMs, such as SWEET (Lee et al., 2023), are designed for code generation. Due to the inherently lower entropy nature in code generation tasks, our method does not design for such scenario. Additionally, this work primarily focuses on the attack. In our future work, we will further investigate how to modify watermarking algorithms to effectively defend against our proposed attack.

**Theoretical**: This work primarily focuses on empirical research. Our attack method targets phrase-level attacks across different watermarking techniques for LLMs and relies on an empirical results, similar to DIPPER. Due to the varying assumptions underlying different watermarking techniques, we cannot provide a theoretical guarantee proof for the final attack's success on each watermark type. Meanwhile establish the upper or lower bound on its performance is not in the scope of this work.

## CONTENTS OF THE APPENDIX

The contents of the supplementary material are organized as follows:

- In appendix A, we list the hyperparameters of the watermarking algorithm we used in experiment section 4.
- In appendix B, w we present the specific data points corresponding to the figure shown in section 4.2.
- In appendix C, we provide the precise data underlying the figure depicted in section 4.3.
- In appendix D, we provide the prompt we used to generate attack text.
- In appendix E, we conduct extensive experiment to evaluate the overall preservation of the semantic meaning of the original watermarked text.
- In appendix F, we offer a brief discussion about the change of self-information under watermark algorithm influence.
- In appendix G, we provide a visual comparison of the text generated by our method with watermarked text, non-watermarked text, and text generated by other attack methods.
- In appendix H, we discuss several watermark attack methods we do not include in our main paper due to setting different.

# A  WATERMARK ALGORITHM SETTING

In this section, we list the hyperparameters of the watermarking algorithm we used in section 4 below.

```
{
    "algorithm_name": "KGW",
    "gamma": 0.5,
    "delta": 2.0,
    "hash_key": 15485863,
    "prefix_length": 1,
    "z_threshold": 4.0
}
```

Listing 1: configuration KGW

```
{
    "algorithm_name": "Unigram",
    "gamma": 0.5,
    "delta": 2.0,
    "hash_key": 15485863,
    "z_threshold": 4.0
}
```

Listing 2: configuration Unigram

```
{
    "algorithm_name": "UPV",
    "gamma": 0.5,
    "delta": 2.0,
    "z_threshold": 4.0,
    "prefix_length": 1,
    "bit_number": 16,
    "sigma": 0.01,
    "default_top_k": 20,
}
```

Listing 3: configuration UPV

```
{
    "algorithm_name": "EWD",
    "gamma": 0.5,
    "delta": 2.0,
    "hash_key": 15485863,
    "prefix_length": 1,
    "z_threshold": 4.0
}
```

Listing 4: configuration EWD

```
{
    "algorithm_name": "DIP",
    "gamma": 0.5,
    "alpha":0.45,
    "hash_key": 42,
    "prefix_length": 5,
    "z_threshold": 1.513,
    "ignore_history": 1
}
```

Listing 5: configuration DIP

```
1  {
2      "algorithm_name": "SIR",
3      "delta": 1.0,
4      "chunk_length": 10,
5      "scale_dimension": 300,
6      "z_threshold": 0.0,
7  }
```

Listing 6: configuration SIR

```
1  {
2      "algorithm_name": "EXP",
3      "prefix_length": 4,
4      "hash_key": 15485863,
5      "threshold": 2.0,
6      "sequence_length": 230
7  }
```

Listing 7: configuration EXP

## B BEST F1 SCORE AND TPR/FPR

In table 6, we list the specific data from the figure in fig. 3, which reflects different attack method's performance of dynamically adjusting the watermark detector's z-threshold until a specified false positive rate is achieved. We report both the F1 score and true positive rate. It can be observed that, in most cases, our attack method achieves the best performance.

Table 6: In this experiment, we dynamically adjust the z-score threshold of the watermarking algorithm until achieving specified false positive rates for the watermark detector. We display the true positive rate and F1 score at this threshold. We also present the classifier's optimal F1 score under the dynamic threshold settings. For attack methods, lower TPR and F1 scores at a given FPR indicate that the watermark detector struggles more to differentiate between attack texts and human-written texts, suggesting a more effective attack.

| Method | Attack Type | 1% FPR | | 10% FPR | | Best F1 (%) |
|---|---|---|---|---|---|---|
| | | TPR (%) | F1 (%) | TPR (%) | F1 (%) | F1 (%) |
| **KGW-1** | No attack | 100 | 99.5 | 100 | 95.2 | 99.8 |
| | DIPPER -1 | 1.6 | 3.1 | 7.8 | 13.3 | **66.6** |
| | DIPPER -2 | **0.8** | **1.6** | **7.0** | **12.1** | **66.6** |
| | GPT Paraphraser | 1.8 | 3.6 | 8.7 | 14.0 | 66.8 |
| | SIRA | 1.1 | 2.1 | 8.4 | 14.0 | **66.6** |
| **Unigram** | No attack | 100 | 99.5 | 100 | 95.2 | 99.8 |
| | DIPPER -1 | 89.0 | 93.7 | 97.6 | 95.8 | 95.4 |
| | DIPPER -2 | 86.0 | 91.8 | 96.8 | 94.2 | 94.6 |
| | GPT Paraphraser | 73.6 | 84.3 | 91.7 | 91.6 | 91.6 |
| | SIRA | **29.0** | **44.6** | **62.2** | **72.7** | **80.0** |
| **UPV** | No attack | 100 | 99.5 | 100 | 95.2 | 99.8 |
| | DIPPER -1 | 74.8 | 85.2 | 94.4 | 93.1 | 93.1 |
| | DIPPER -2 | 67.8 | 80.4 | 91.6 | 91.6 | 91.6 |
| | GPT Paraphraser | 53.0 | 69.2 | 82.2 | 86.2 | 86.9 |
| | SIRA | **27.0** | **42.3** | **65.6** | **75.4** | **81.4** |
| **EWD** | No attack | 100 | 99.5 | 100 | 95.2 | 99.8 |
| | DIPPER -1 | 66.2 | 79.2 | 91.4 | 90.8 | 90.8 |
| | DIPPER -2 | 54.8 | 70.3 | 83.2 | 81.2 | 81.2 |
| | GPT Paraphraser | 46.6 | 63.1 | 75.4 | 81.3 | 83.8 |
| | SIRA | **10.2** | **18.3** | **35.8** | **49.1** | **71.6** |
| **DIP** | No attack | 100 | 99.5 | 100 | 95.2 | 99.8 |
| | DIPPER -1 | 5.4 | 10.0 | 24.0 | 36.1 | 67.7 |
| | DIPPER-2 | 2.2 | 4.3 | 16.4 | 26.2 | **66.7** |
| | GPT Paraphraser | 4.3 | 8.3 | 20.4 | 32.0 | 76.6 |
| | SIRA | **1.6** | **3.1** | **11.2** | **18.7** | **66.7** |
| **SIR** | No attack | 100 | 99.5 | 100 | 95.2 | 99.8 |
| | DIPPER -1 | 64.6 | 78.0 | 82.4 | 85.6 | 85.6 |
| | DIPPER -2 | 57.6 | 72.6 | 72.6 | 83.4 | 83.4 |
| | GPT Paraphraser | 66.2 | 79.2 | 85.2 | 87.3 | 87.3 |
| | SIRA | **42.8** | **59.5** | **70.2** | **77.9** | **82.4** |
| **EXP** | No attack | 100 | 99.5 | 100 | 95.2 | 99.8 |
| | DIPPER -1 | 0.8 | 1.6 | 1.2 | 2.1 | 66.6 |
| | DIPPER -2 | 0.4 | 0.8 | **2.0** | **3.8** | 66.7 |
| | GPT Paraphraser | 0.4 | 0.8 | **2.0** | **3.8** | 82.6 |
| | SIRA | **0** | **5.4** | 9.3 | 8.3 | **66.6** |

## C DETAIL NUMBER OF PPL AND SENTENCE BERT SCORE

In this section, we list the detail number of PPL and sentence-bert score we present in the section 4.3.

| Watermark | KGW-1 | | Unigram | | UPV | | EWD | | DIP | | SIR | | EXP | |
| Attack | PPL(↓) | s-BERT(↑) | PPL(↓) | s-BERT(↑) | PPL(↓) | s-BERT(↑) | PPL(↓) | s-BERT(↑) | PPL | s-BERT(↑) | PPL(↓) | s-BERT(↑) | PPL(↓) | s-BERT(↑) |
|---|---|---|---|---|---|---|---|---|---|---|---|---|---|---|
| No attack | 12.00 | 0.55 | 11.49 | 0.56 | 9.27 | 0.56 | 11.64 | 0.57 | 10.60 | 0.57 | 11.76 | 0.56 | 16.48 | 0.54 |
| Word delete | NaN | 0.03 | NaN | 0.03 | NaN | 0.03 | NaN | 0.01 | NaN | 0.04 | NaN | 0.03 | NaN | 0.04 |
| Synonym Substitution | 252.85 | 0.01 | 252.85 | 0.01 | 252.85 | 0.01 | 252.85 | 0 | 252.85 | 0.01 | 252.85 | 0.02 | 252.85 | 0.02 |
| GPT Paraphraser | 9.19 | 0.88 | 8.96 | 0.87 | 8.28 | 0.87 | 9.20 | 0.88 | 8.79 | 0.88 | 9.52 | 0.87 | 11.98 | 0.86 |
| DIPPER-1 | 12.00 | 0.80 | 11.80 | 0.79 | 10.31 | 0.81 | 11.87 | 0.76 | 11.93 | 0.80 | 12.43 | 0.79 | 16.56 | 0.77 |
| DIPPER-2 | 12.15 | 0.79 | 11.80 | 0.79 | 10.34 | 0.80 | 11.96 | 0.76 | 11.86 | 0.79 | 12.42 | 0.78 | 16.45 | 0.76 |
| SIRA (Ours) | 10.59 | 0.78 | 9.37 | 0.78 | 8.91 | 0.77 | 9.10 | 0.78 | 9.02 | 0.77 | 9.74 | 0.78 | 10.93 | 0.75 |

Table 7: Performance of different attack methods against various watermark techniques.

## D FILL IN BLANK INSTRUCTION

We present our generation prompts below. During the generation process, the reference text and the masked text are replaced by the two texts obtained in the first step.

> **Generation Prompts**
>
> You will be shown one reference paragraph and one incomplete paragraph. Your task is to write a complete paragraph using incomplete paragraph. The complete paragraph should have similar length with reference paragraph. You need to include all the information in the reference. but do not take the expression and words in the reference paragraph. You should only answer the complete paragraph.
> reference: [reference text]
> incomplete paragraph: [mask text]

## E GPT SEMANTIC JUDGE

> **GPT Semantic Judge Prompts**
>
> You are a helpful assistant. You will receive a original text and a pharased text .Please act as an impartial judge and evaluate how much the pharased text keep the original text semantic meaning. Be as objective as possible. Rate the response on a scale from 1 to 10. A rating of 1 signifies the pharased text has totally different meaning with the original text. A rating of 10 signifies the pharased text has same meaning with the original text . Your answer should strictly be the format: [Rating].
> Here's the original text: [Watermarked text]
> Here's the pharsed text:[Attack text]

In this section, we conduct extensive experiments to evaluate the overall preservation of the semantic meaning of the original watermarked text. We use ChatGPT (OpenAI, 2024) as an impartial judge to obtain the quantitative results.

The attack success rate alone is not a sufficient metric for evaluating an attack method. It is also crucial to assess whether the original and paraphrased outputs preserve similar semantics. The Sentence-BERT score (Reimers & Gurevych, 2019), presented in section 4.3 , measures the sentence-level similarity between the original watermarked text and the adversarial text. However, it falls short in determining whether the overall semantics are preserved. Inspired by the LLM jailbreak work PAIR (Chao et al., 2023), which leverages carefully crafted prompts and the powerful capabilities of ChatGPT to score attack texts and targets for quantitative evaluation, we adapted their prompts to use ChatGPT for assessing the semantic similarity between watermarked texts and attack texts . This approach allows us to obtain semantic similarity scores that more closely align with human perception. We show the judge prompt in appendix E and the result in shown in table 8.

We observed that using GPT for paraphrasing alone best preserves the original text's semantics, whereas methods like word deletion and synonym replacement were largely ineffective. Our ap-

Table 8: Semantic Preservation for Different Methods

| | Word delete | Synonym | GPT Paraphraser | DIPPER-1 | DIPPER-2 | SIRA |
|---|---|---|---|---|---|---|
| **Semantic Preservation** | 2.59 | 2.63 | 8.25 | 5.28 | 6.34 | 6.84 |

proach demonstrated superior semantic preservation compared to the DIPPER method. Additionally, we conducted experiments using LLaMA3-70B and found that, due to the model's enhanced capability to comprehend the "preserve information" instruction in the rewrite task, the semantic preservation score increased to 7.34. Furthermore, there was a 10%-20% reduction in perplexity, depending on the specific watermarking algorithm, and 10% higher attack success rate, when LLaMA3-70B was used as the base model.

## F  SELF-INFORMATION, ENTROPY AND PROBABILITY

We provide a brief explanation of how watermark algorithm change the self-information. To begin, we introduce the definitions of self-information.

**Self-Information** ($I(x)$): This measures the amount of information or "surprise" associated with a specific token $x$. It quantifies how unexpected the occurrence of a token is in a given context:

$$I(x) = -\log_2 P(x)$$

When considering the context $h$, it becomes the conditional self-information:

$$I(x \mid h) = -\log_2 P(x \mid h)$$

where $P(x \mid h)$ is the probability of token $x$ occurring given the preceding context $h$.

We first analyze the non-conditional scenario, assuming that watermarking slightly increases the probability of certain tokens by a small amount $\delta$, while adjusting the probabilities of other tokens to maintain normalization. The $\delta$ change in token influcenced by watermark algorithm usually very small (e.g less than 1e-3).

The adjusted probability for the watermarked token $x_w$ is:

$$P'(x_w) = P(x_w) + \delta$$

The adjusted probabilities for other tokens $x_i$ ($i \neq w$) are:

$$P'(x_i) = P(x_i) - \epsilon_i$$

where $\sum_{i \neq w} \epsilon_i = \delta$.

The change in entropy due to these adjustments is given by:

$$\Delta H = H(P') - H(P) = -\sum_i \left[ P'(x_i) \log P'(x_i) - P(x_i) \log P(x_i) \right]$$

The partial derivative of entropy with respect to $P(x_w)$ is:

$$\frac{\partial H}{\partial P(x_w)} = -\log P(x_w) - 1$$

The change in entropy due to a small change $\delta$ in $P(x_w)$ is approximately:

$$\Delta H \approx \frac{\partial H}{\partial P(x_w)} \delta = -(\log P(x_w) + 1)\delta$$

In high-entropy contexts, where $P(x_w)$ is small, $\log P(x_w)$ becomes a large negative value. Therefore, $\log P(x_w) + 1$ is still negative, and the product with the small $\delta$ results in a tiny $\Delta H$(decrease in logarithmically). This attribute makes watermark algorithm need to embed pattern in high-entropy tokens, otherwise it will significantly compromising the quality of the output.

For self-information, the change in self-information is:

$$\Delta I(x_w) = -\log P'(x_w) + \log P(x_w)$$

The derivative of self-information with respect to $P(x_w)$ is:

$$\frac{dI(x_w)}{dP(x_w)} = -\frac{1}{P(x_w)}$$

For small $P(x_w)$, $\frac{1}{P(x_w)}$ is large, making $\Delta I(x_w)$ more significant for small $\delta$ compared to $\Delta H$.

Similarly, for conditional self-information, assume that the model predicts $N$ possible next tokens with equal probability, where:

$$P(x \mid \text{Context}) = \frac{1}{N}$$

For large $N$, $P(x \mid \text{Context})$ becomes small.

The adjusted probability for the watermarked token $x_w$ is:

$$P'(x_w \mid \text{Context}) = \frac{1}{N} + \delta$$

The adjusted probabilities for other tokens $x_i$ ($i \neq w$) are:

$$P'(x_i \mid \text{Context}) = \frac{1}{N} - \frac{\delta}{N-1}, \quad \text{for } x_i \neq x_w$$

The change in Conditional Self-Information is:

$$\Delta I(x_w \mid \text{Context}) = I'(x_w \mid \text{Context}) - I(x_w \mid \text{Context}) = -\log\left(\frac{1}{N} + \delta\right) + \log N$$

Using a Taylor series approximation for small $\delta$:

$$\log\left(\frac{1}{N} + \delta\right) \approx \log\left(\frac{1}{N}\right) + N\delta$$

The approximate change in conditional self-information is:

$$\Delta I(x_w \mid \text{Context}) \approx -N\delta$$

Compared to the change in entropy, it is obvious self-information are more sensitive metric:

$$\Delta H \approx \frac{\partial H}{\partial P(x_w)}\delta = -(\log P(x_w) + 1)\delta$$

When $P(x \mid \text{Context})$ is small, the magnitude of the derivative is large, this indicates that small changes in $P(x \mid \text{Context})$ result in bigger changes in $I(x \mid \text{Context})$. As a result, the green token influenced by watermark change will have less self-information than original.

High-entropy tokens are usually associated with high self-information due to their unpredictability and low probability of occurrence. Considering the reduce self-information, these potential green tokens generally will exhibit high or moderate self-information values. Therefore in practice, we filter out all tokens with high or moderate self-information. This ensures us can comprehensive eliminate potential tokens. As demonstrated in our experiments in table 4, a threshold of 0.4 achieves an 88% attack success rate.

Another advantage of self-information is that it is **computed for each token within its specific context**, allowing it to naturally adapt to the varying nature of different sequence types. This provides a dynamic and context-sensitive measure that better aligns with the structure of natural sequences. In contrast, **entropy is context-agnostic, treating all token sequences equally when calculating average uncertainty**. By leveraging self-information, which adapts to each token's context, it becomes easier to identify sequences that deviate from expected contextual patterns — such as those that may be watermarked. We believe this is a key reason why filtering by self-information empirically outperforms filtering by entropy.

## G  VISUALIZATION

In this section, we present a visual comparison of our algorithm with other model-based paraphrasing methods, along with the corresponding z-scores after the attack. For discrete methods, green tokens are marked in green, and red tokens in red. In the watermarking algorithm, the detector identifies the embedded watermark through green tokens and calculates the z-score; fewer green tokens or a lower z-score indicate a more successful attack. For continuous methods, the shade of color denotes the weight of the watermarked token, with darker colors representing higher weights. In the case of attacked text, lighter colors indicate a more successful attack.

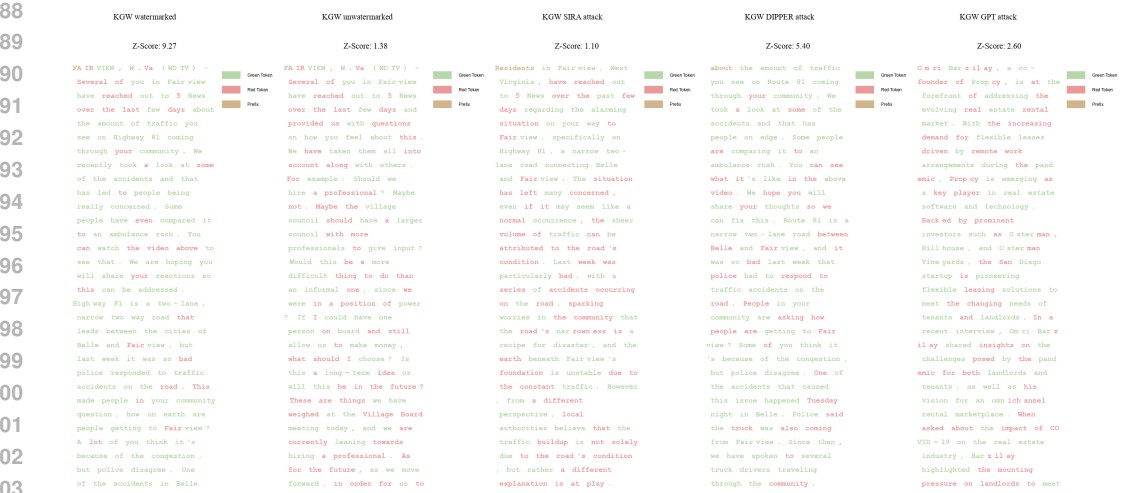

Figure 5: Comparison of different paraphrasing methods on KGW watermarks. Each word's color indicates whether it is a green or red token. **Fewer green words/lower z-scores** suggest a more effective paraphrasing approach. The unwatermarked text represents the model's output without the influence of the watermarking algorithm. The example demonstrates that our method achieves a better z-score than the unwatermarked text..

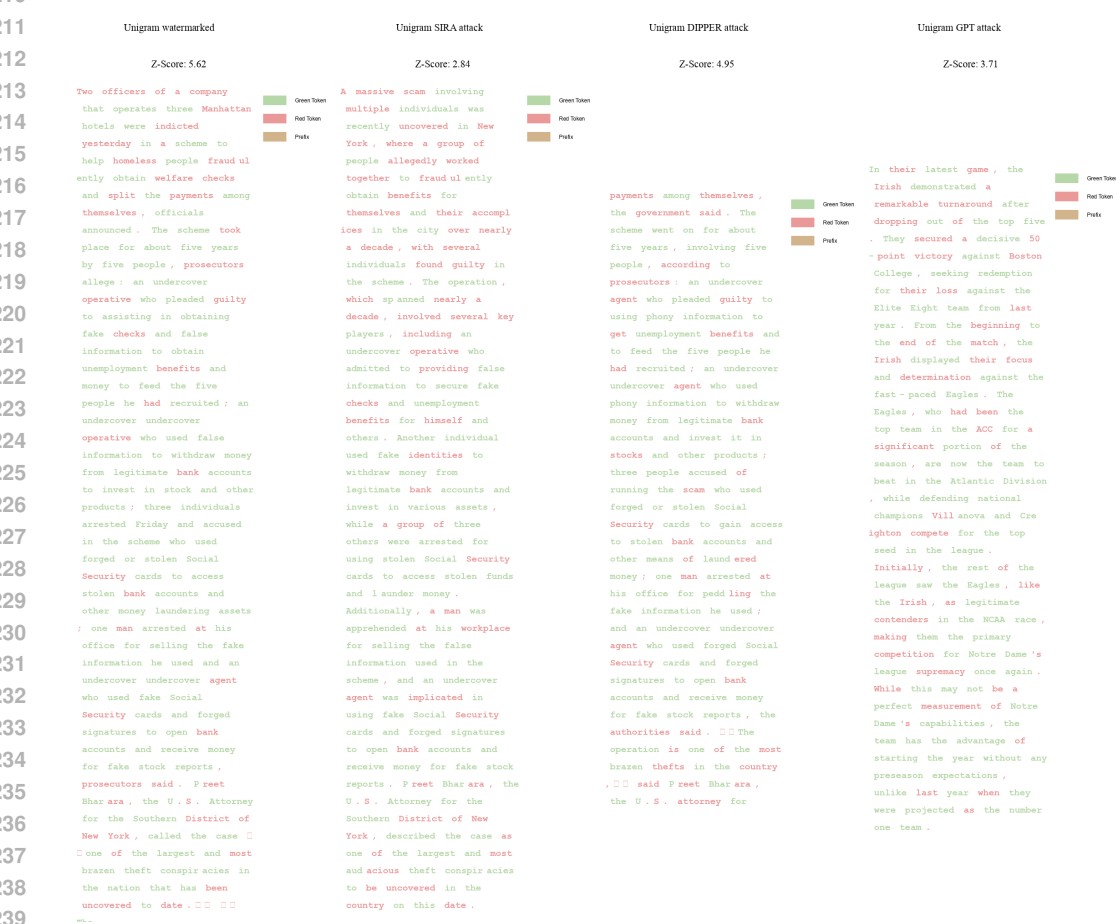

Figure 6: Comparison of different paraphrasing methods on Unigram watermarks.

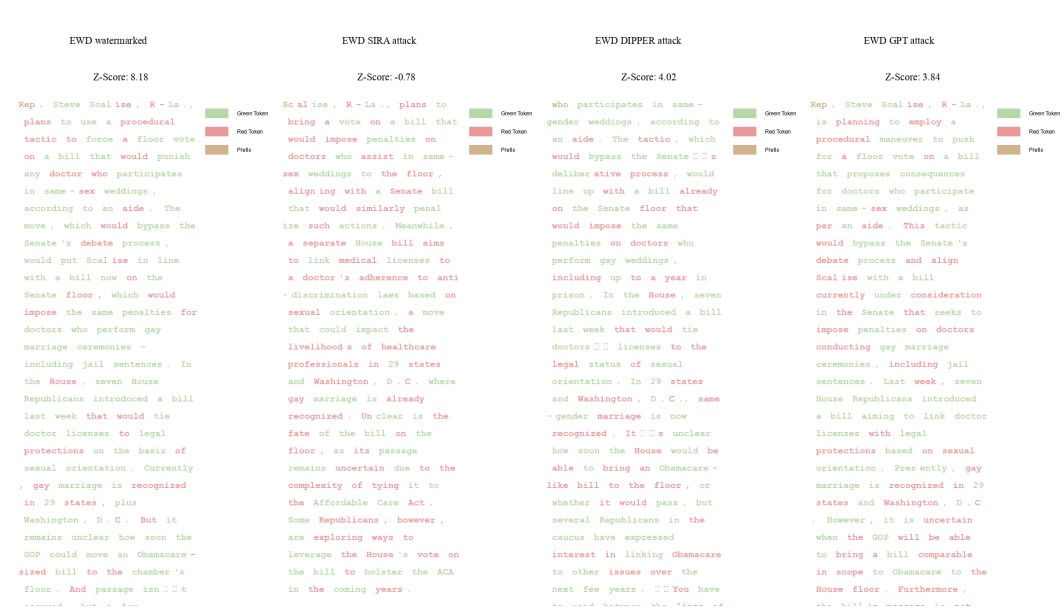

Figure 7: Comparison of different paraphrasing methods on EWD watermarks.

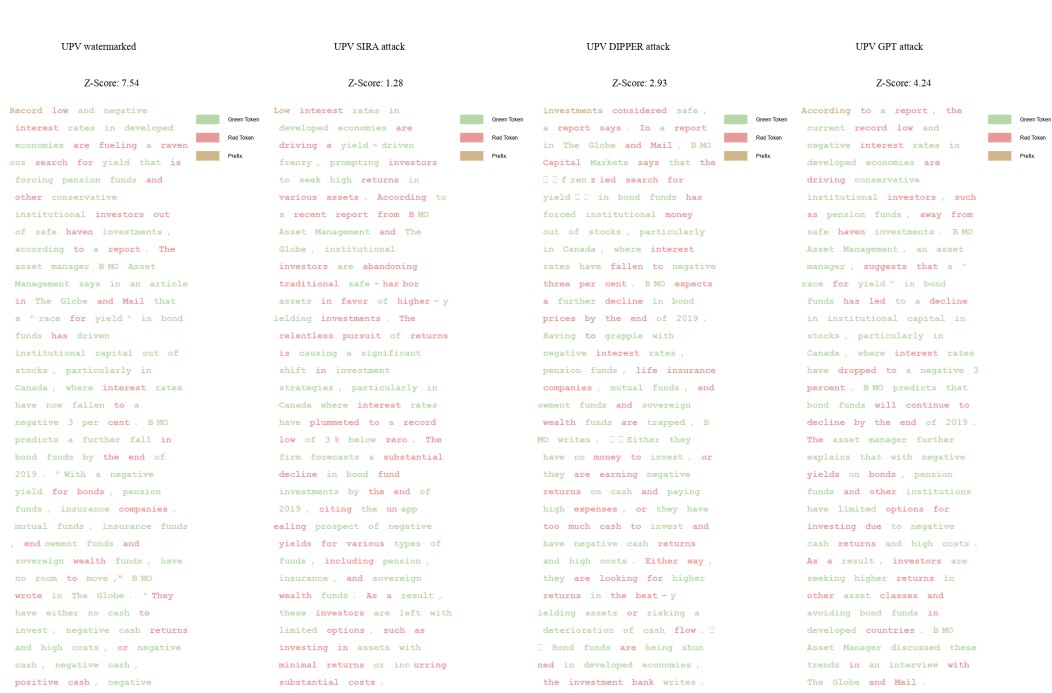

Figure 8: Comparison of different paraphrasing methods on UPV watermarks. The color of each word indicates whether it belongs to a green token or a red token. **Less green** signifies a more effective paraphrasing approach. Our methods show better performance in removing original watermark text green token.

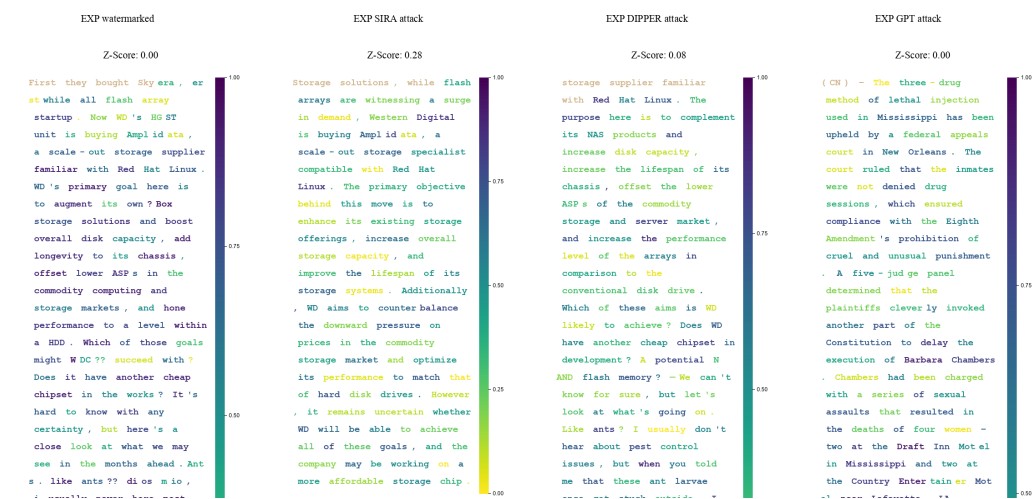

Figure 9: Comparison of different paraphrasing methods on EXP watermarks. The color of each word indicates whether it is a green or red token. For EXP, **lighter word colors and higher z-scores indicate a more effective attack**.

# H    EXTENDED DISCUSSION IN RELATED WORKS

In this section, we discuss the watermark-stealing attack (Jovanović et al., 2024; Wu & Chandrasekaran, 2024; Zhang et al., 2024), which is related to our work but operates under different assumptions. We give a brief introduction to this kind of attack and analyze its advantages and disadvantages compared to our methods.

In a typical watermark-stealing attack, an adversary interacts with a watermarked LLM model through its API or queries, attempting to reverse-engineer or approximate the embedded watermark. Through repeated queries, attackers may identify patterns in token choices or subtle markers in output text that are indicative of a watermark. Two main attack methods emerge from this process:

- Spoofing Attacks: Here, attackers use the approximated watermark pattern to make non-AI-generated text appear as though it was produced by the LLM, thereby deceiving detection systems.
- Scrubbing Attacks: These attacks aim to remove the watermark from AI-generated text. By understanding how the watermark is embedded, attackers substitute or adjust specific tokens or structures to evade detection, making AI-generated content appear as if it is human-authored.

For watermark-stealing attacks, these works assume that the attacker has:

1. Unlimited access to the watermark-generated model API, through which the attacker can issue input prompts and observe the generated responses.
2. Access to the detector API (with or without different assumptions).
3. Knowledge of the context size.
4. An aligned watermarked model that can follow provided instructions.

In a black-box paraphrasing attack, we assume that the attacker's knowledge is limited to only the watermarked text and nothing else. Because this assumption is much weaker than that of a watermark-stealing attack, we do not include such methods in our main comparisons.

Compared to the paraphrasing attack, these methods have the following advantages:

1. Minimal Semantic Drift: By focusing on substituting only the watermarked tokens, such methods tend to retain the original semantic structure of the text more effectively than black-box rewriting, which can sometimes introduce unintended semantic changes.

2. Fine-Grained Control: Watermark-stealing attacks operate at a token level, allowing precise control over watermark removal without needing to rephrase entire sentences or sections, which may be challenging in highly sensitive content.

3. Better piror knowledge: Since such attacks assume the attacker has unlimited access to the watermarked model, they can continuously probe the watermark pattern through specific input queries, thereby gaining stronger prior knowledge.

These methods also have disadvantages:

1. Possible Efficiency Issue: As mentioned in SCTS (Wu & Chandrasekaran, 2024), repeated probing introduces additional time overhead, resulting in higher resource consumption compared to paraphrasing methods.

2. Limited Scope of Application: Black-box rewrite attacks do not require knowledge of the watermarking mechanism, making them suitable for use against a wide variety of watermarking techniques.

3. Possible lower robustness: Because black-box rewrite attacks focus on altering the content through paraphrasing, they can reduce the chance of detection by watermarking systems that rely on token-level patterns.

Notably, we need to mention SCTS (Wu & Chandrasekaran, 2024). This method also involves modifying color tokens, but it is entirely different from our proposed approach. In SCTS, The attacker prompts the watermarked LLM multiple times with various inputs, collecting a significant amount of output data to capture the frequency distribution of tokens. By examining this collection of outputs, the attacker can calculate the frequency of each token appearing in the generated text. Watermarked LLMs favor certain tokens, so these "green" tokens appear more frequently than others due to the watermarking bias. The attacker observes these frequencies for anomalies or patterns that deviate from what would be expected in unwatermarked text. Tokens that consistently appear at higher frequencies or in specific contexts are likely candidates for "green" tokens.

Therefore, this method identifies green tokens using a frequency-based approach and has two main limitations. First, it shares a common limitation of such methods, requiring access to the LLM API and permission to adjust its parameters. Second, it is only applicable to biased watermarking methods, such as UMD and its variants.

In contrast, the key insight of our method is that current watermarking techniques require embedding on high-entropy or uniformly distributed tokens to maintain text quality, as explained in Section F. By leveraging self-information, we can exploit this characteristic to identify potential green tokens and rewrite them accordingly. Our method offers greater flexibility as it does not require access to the watermarking model and not limited to biased watermark algorithm.

