# OpenReview forum: "SIRA: Exposing Vulnerabilities in Text Watermarking with Self-Information Rewrite Attacks"
_ICLR.cc/2025/Conference — ICLR 2025 Conference Withdrawn Submission_

### Official Review · Reviewer_ATEL · 2024-11-02

**Soundness:** 1
**Presentation:** 3
**Contribution:** 2
**Rating:** 3
**Confidence:** 4

**Summary:**

This is an interesting paper, it introduces the Self-Information Rewrite Attack (SIRA) as a novel watermark removal method targeting the robustness of text watermarking techniques used in content generated by large language models (LLMs). These watermarks allow tracing and verifying content origins to prevent misuse.

**Strengths:**

The proposed method, namely SIRA, leverages the high self-information of certain tokens where watermarks are often embedded, applying a lightweight masking and rewriting technique to evade watermark detection. The experimental results demonstrate that SIRA effectively disrupts watermarking in seven algorithms with minimal computational resources, achieving a 90% success rate, suggesting critical vulnerabilities in current watermarking techniques.

**Weaknesses:**

1. **Unclear motivation.** In lines 57-59, the authors point out the limitations of previous methods and later propose their method to overcome these challenges. One limitation is transparency, but why is transparency desirable for watermark removal attacks via paraphrasing? The second limitation is computational cost, but their method requires three models: one for masking, one to generate reference text, and a third one for paraphrasing given the masked reference text, which is not computationally efficient.

2. **The method lacks soundness.** How can it be ensured that the LLaMA 3 rewriting can favor red tokens to achieve watermark removal? Why can this method remove SIR watermarking while existing methods fail?

3. **Unclear experiment settings.** Which models are used as $ M_{\text{attack}} $ (line 290) and the base LLM (line 307), respectively? How does LLaMA3-8b ensure the lightweight and usability of the proposed method? The lightweight aspect of LLaMA3-8b is questionable, considering the watermarked model in the experiment is OPT 1.3B.

4. **The experimental results lack analysis and sometimes are even inconsistent.** Why does word deletion yield a high s-BERT score for KGW-1 (i.e., deletion preserves sentence-level embedding similarity) but low scores for other watermarks in Figure 4b? Why are the results in Figure 4b inconsistent with those in Table 8?

5. **Minor issues:**
   - Typo: `paraphase` in Figure 1 and its caption.
   - The threshold in line 296 should be $\epsilon$ instead of $\sigma$.

**Questions:**

I have listed most of my questions associated with these weaknesses above.

---

> ### Author Response · Authors · 2024-11-19
> **rebuttal part 1**
>
> We thank reviewer ATEL for the valuable time for providing us feedback. We address the concern below:
>
>
>  > **Q1**: motivation and why is transparency desirable for watermark removal attacks via paraphrasing?
>
> The need for a powerful, lightweight tool to evaluate watermark robustness is a well-recognized problem in text watermarking, as noted by reviewer 91eq. Current paraphrasing-based watermark removal methods like GPT Paraphraser operate as black boxes. This opacity means that repeated attempts to attack the same watermarked text may produce highly variable results due to random variations in the black box paraphrasing process.
>
> In contrast, our approach is grounded in the well-established concept of self-information. The generated mask template is deterministic and targeted, which reduces randomness and instability, thereby enhancing the method's effectiveness and stability.
>
>
> > **Q2** limitation is computational cost, three model are used
>
> A2:  **We clarify here only one Llama3-8b model is used in our method as we mentioned in Lines 69, 350, 461**. The total time consumption for SIRA consists of two parts: two paraphrasing and self-information mask.The self-information mask is nearly negligible, as it does not require any text generation (less than 0.1 seconds).  The other two generations take around 4-5 seconds per generation on a single A100 GPU. Thus the total execution time is around 10 seconds. We use huggingface transformer in our experiment，the presented results could easily be validated by several lines of code.
>
> In comparison, the DIPPER method utilizes a specially fine-tuned model for text paraphrasing. Its larger parameter size (indicated by vram in table 2) and model architecture result in *15 seconds per run on two A100 GPUs*. Notably, DIPPER inherently relies on this fine-tuned model, preventing it from transferring to a smaller model. Consequently, our method is faster and has weaker resource requirements.
>
>
> > **Q3**  How can it be ensured that the LLaMA 3 rewriting can favor red tokens to achieve watermark removal?
>
> A3:  For a well-designed watermarking algorithm, the probability of identifying text generated by an LLM (without watermarking intervention) as positive should theoretically be zero. Otherwise the watermark algorithm is problematic which will have a high false positive rate.
>
> The reason why previous paraphrasing approaches fall short in bypassing watermark detection is
> simple paraphrasing  is generated with context (original watermarked text), with such context LLM tend to preserve original context component (e.g words, expression), so even new generated content part are “clean” and will dilute the strength of watermark, **the leaving behind n-gram remnants from original text that still allow detectors to identify the watermark pattern**. By contrast, **our method creates a cleaner template that actively removes potential green tokens,  minimizing n-gram remnants** and thereby significantly enhancing the attack success rate as we showed in the experiment part.
>
> > **Q4**Why can this method remove SIR watermarking while existing methods fail?
>
> A4: We respectfully disagree with the reviewer as **we never make such a claim in our paper**.  Our method outperforms other attack baseline in SIR watermark but we do not claim existing methods failed. Notably, we are only 1.4% better than synonyms substitutes. SIR dynamically produces a green list based on the preceding tokens to the watermark logits. Our proposed self-information is also based on the conditional of preceding tokens. Meanwhile, The SIR watermark logit exclusively 1 or -1, this makes token self-Information value change almost the same. This results in most SIR-embedded token self-information values falling within the same range, making our percentile-filter highly effective.
>
> > **Q5** Which models are used as (line 290) and the base LLM (line 307), respectively?
>
> A5: We clarify here only one model is used in our method. We kindly remind the reviewer we already explain in Line 289 that the base LLM refers to M attack. We thank the reviewer's suggestion and will change accordingly to reduce possible misunderstanding.

---

> > ### Author Response · Authors · 2024-11-19
> > **rebuttal part 2**
> >
> > > **Q6**    How does LLaMA3-8b ensure the lightweight and usability of the proposed method?
> >
> > A6: We have listed the vram consumption and execution time in sec 4.4. The experiment empirically shows our proposed method consumes less resources while achieving a high attack success rate. DIPPER relies on the specific fine-tune model thus can not transfer to smaller model, our method *can work on the light-weight model* which requires half VRAM and achieve better attack performance as acknowledged by reviewer EJ9Y.
> >
> > > **Q7** The lightweight aspect of LLaMA3-8b is questionable, considering the watermarked model in the experiment is OPT 1.3B.
> >
> > A7: We do not understand the reviewer's comment.  OPT1.3B is a **common practice** model used in watermark works [1,2,3,4]. **The strength and robustness of the watermark mostly decided by the design of the watermarking algorithm and its hyperparameter**. Specifically, Unigram[2]  adopts the same setting with ours which uses opt-1.3b to generate watermarked text and uses a larger DIPPER model to execute attacks to test watermark robustness.
> >
> > Moreover, **the comparison of methods should be made against baseline attack methods**, such as the DIPPER,GPT model, with our method, rather than the tool model to generate watermark text.  We would be grateful if the reviewer further explains why such a comparison should be made and provide any reference papers that have conducted comparisons in the manner suggested.
> >
> > > **Q8** The experimental results  are  inconsistent.
> >
> > A8: We have corrected and updated the wrong data in our revised manuscript for KGW-1 word deletion. We kindly remind reviewer the Figure 4(b) and Table 8 are **two different experiment with different metrics**. Figure 4b uses s-bert score to evaluate sentence-level, while Table 8 uses ChatGPT as a judge to evaluate overall semantic preservation.
> >
> > > **Q9** Typos
> >
> > A9: We are thankful for the reviewer's correction. We have fixed the typo in our revised manuscript.

---

> > > ### Author Response · Authors · 2024-11-19
> > > **rebuttal reference**
> > >
> > > 1] Krishna, K., Song, Y., Karpinska, M., Wieting, J., & Iyyer, M. “Paraphrasing evades detectors of ai-generated text, but retrieval is an effective defense.” NeruIPS , 2024
> > >
> > > [2] Zhao, X., P. Ananth, L. Li, and Y.-X. Wang. "Provable Robust Watermarking for AI-Generated Text." ICLR, 2024.
> > >
> > > [3] Pan, L., A. Liu, Z. He, Z. Gao, X. Zhao, Y. Lu, B. Zhou, and S. Liu. "MarkLLM: An Open-Source Toolkit for LLM Watermarking." EMNLP, 2024.
> > >
> > > [4] Kuditipudi, R., J. Thickstun, T. Hashimoto, and P. Liang. "Robust Distortion-Free Watermarks for Language Models." TMLR, 2023.

---

> ### Author Response · Authors · 2024-11-22
> **Are the concerns of the Reviewer ATEL addressed?**
>
> Dear reviewer ATEL,
>
> We appreciate your time for providing us feedback. We strive to address your questions in our previous responses. We strive to address your questions in our previous responses.  The key concerns were motivation, the mechanism of LLMs watermark,  the possible misunderstanding regarding experiment setting and experiment results. We also clarify that we only use one model in our method and the mentioned two inconsistent results are different experiments. Our response above addresses those concerns. If there are specific parts that are unclear to the reviewer, we are more than happy to revise them.
>
> Please let us know if you have any remaining concerns.We thank you again for your valuable time and feedback.
>
>
> Best,
> Authors

---

> ### Author Response · Authors · 2024-12-02
> **Did our response address the reviewer’s concerns?**
>
> Dear Reviewer ATEL,
>
> Thank you for your valuable feedback and the time you've devoted to reviewing our work. With the discussion period will end in 24 hours, we note that we have not yet received any further comments from you.
>
> If you feel your concerns have been adequately addressed, we kindly ask you to consider revising your score. If you have any remaining concerns, we would be glad to provide additional clarifications.
>
> Best regards, The Authors

---

### Official Review · Reviewer_AqqW · 2024-11-02

**Soundness:** 3
**Presentation:** 2
**Contribution:** 2
**Rating:** 5
**Confidence:** 3

**Summary:**

The paper proposes a model-based paraphrasing attack. It identifies potential green words and provides a template for rephrasing.

**Strengths:**

The attack is performed in a black-box setting.

The semantics of the paraphrased text is preserved.

**Weaknesses:**

The paper has limited optimization for preserving semantics. The primary strategy for maintaining semantics is changing only potential "green" words while providing a masked template. However, the paper could be strengthened by presenting evidence on how the template contributes to semantic preservation.

Additionally, the method requires two paraphrasing steps: one to generate a reference text and another to create the attack text. The paper would benefit from explaining how it achieves shorter execution time.

In the ablation study, besides comparing the self-information mask with the random mask, it would be valuable to include a comparison between the self-information mask and no mask (paraphrasing twice).

**Questions:**

Why does SIRA have a shorter execution time than GPT Paraphraser, considering that SIRA requires two paraphrasing steps? Table 2 indicates that the execution speed of GPT Paraphraser may vary depending on the network status and real-time OpenAI server load. Does this make for a fair comparison?

---

> ### Author Response · Authors · 2024-11-19
> **rebuttal part 1**
>
> We sincerely thank the reviewer for the comments and constructive suggestions. We address the concerns below:
>
>  > **Q1**: The primary strategy for maintaining semantics is changing only potential "green" words while providing a masked template.
>
> A1: We want to clarify the misunderstanding here. **The primary role of the template is to form the attack, not to preserve semantics**.The reason why previous attacks do not perform as well as us is that the brute force manner paraphrasing tends to preserve original watermark green token components (e.g., words, expression).  The newly generated or changed text will dilute the remnant watermark pattern strength. However, since these methods rely solely on asking the LLM to paraphrase, the process is untargeted and uncontrollable, resulting in remnant watermark patterns that still allow detectors to identify the text as watermarked.  In contrast, **our method providing this “cleaner template” by actively removing potential "green" tokens,  transforms the paraphrasing task into a task more similar to a fill-in-the-blank task to achieve attack**.  Our main experiment presented in Table 1, along with the valuable ablation study requested by the reviewer (addressed in Q4), demonstrates that our template-based approach did lead to better attack performance.
>
>  > **Q2**: Why does SIRA have a shorter execution time?
>
> A2: The total time consumption for SIRA consists of two parts: two generations by the base model and  the self-information mask. The self-information mask is nearly negligible, as it does not require any text generation (less than 0.1 seconds).  The other two generations will run around 4-5 seconds(256 tokens) per generation on a single A100 gpu. Thus the total execution time is around 10 seconds. We use the open-source huggingface library in our experiment, the shown result could be easily validated by a few  lines of code. For details of GPT Paraphraser and DIPPER, please refer to our global response and A3.
>
>  > **Q3**:  Computation Cost compared to GPT Paraphraser not fair
>
> A3:  We emphasize that GPT Paraphraser uses a closed-source model, limiting our access to details. OpenAI primarily uses H100 clusters, according to the OpenAI forum. We can only try our best to ensure a fair and comprehensive comparison, with the note solely aimed at maintaining transparency and rigor for readers.
>
> To fully address the reviewer’s concerns，We retested the execution speed of GPT Paraphraser on the same day with a 4-hour interval for 4 times in one day, conducting a total of 4 (times) * 7 (watermarks) * 50 (queries) = 1400 samples. The final average time per text is 12.6±0.4 seconds which is aligned with the data we showed in paper. We are glad to conduct further tests if the reviewer suggests a more rigorous testing method.
>
> From another perspective, consider the cost of processing 1M tokens of watermark text using third-party services. Using GPT Paraphraser would cost 20 \\$ (input+output), according to OpenAI's price list. In contrast, our method costs 0.22 $\times$ 2(input + output) $\times$ 2 (two iterations) = 0.88 \$, based on AWS Bedrock pricing. Our cost is significantly lower.
>
>
>
>  > **Q4**:  Ablation for paraphrasing twice with no mask
>
> A4: We agree with the reviewer that such an ablation study is necessary to ensure the reliability of the conclusions and thankful for the reviewer's insightful suggestion. We conduct the experiment and show the data in the table below. The experiment follows the ablation part setting and uses Unigram as the watermark algorithm.
>
> |                                       | ASR  |
> | ------------------------------------- | ---- |
> | No mask twice            |  70 |
> | Self-information Mask | 96   |
>
> The results show no mask but twice paraphrasing ASR is 70% which is lower than our proposed methods.

---

> > ### Author Response · Authors · 2024-11-19
> > **rebuttal reference**
> >
> > [1]  Krishna, K., Song, Y., Karpinska, M., Wieting, J., & Iyyer, M. “Paraphrasing evades detectors of ai-generated text, but retrieval is an effective defense.” NeruIPS , 2024

---

> ### Author Response · Authors · 2024-11-22
> **Are the concerns of the Reviewer AqqW addressed?**
>
> Dear reviewer AqqW,
>
> We appreciate your feedback and suggestions on our work. We strive to address your questions in our previous responses. The key concerns were why SIRA has shorter execution time, an ablation comparison with paraphrase twice, and cost comparison with GPT Paraphraser. We also clarify that the mask was not designed to keep semantics but to form an attack. Our response above addresses those concerns and include the new experiment required by the reviewer.  If there are specific parts that are unclear to the reviewer, we are more than happy to revise them.
>
> Please let us know if you have any remaining concerns.We thank you again for your valuable time and constructive feedback.
>
>
> Best,
> Authors

---

> ### Author Response · Authors · 2024-12-02
> **Did our response address the reviewer’s concerns?**
>
> Dear Reviewer AqqW,
>
> Thank you for your valuable feedback and the time you've devoted to reviewing our work. With the discussion period will end in 24 hours, we note that we have not yet received any further comments from you.
>
> If you feel your concerns have been adequately addressed, we kindly ask you to consider revising your score. If you have any remaining concerns, we would be glad to provide additional clarifications.
>
> Best regards, The Authors

---

> ### Comment · Reviewer_AqqW · 2024-12-02
>
> Thank you for the response! I will keep the scores as they are.

---

> ### Author Response · Authors · 2024-12-03
>
> We thank the reviewer for the response! We would greatly appreciate it if the reviewer could let us know whether your concerns have been addressed, as this will help us further improve our paper.

---

### Official Review · Reviewer_91eq · 2024-11-03

**Soundness:** 3
**Presentation:** 3
**Contribution:** 2
**Rating:** 5
**Confidence:** 4

**Summary:**

This paper presents the Self-Information Rewrite Attack (SIRA), a watermark removal method that targets vulnerabilities in existing watermarking techniques applied to LLM-generated text. By using self-information to identify and modify high-entropy tokens, SIRA effectively removes watermarks while preserving text quality. The authors conduct extensive experiments across multiple watermarking schemes and demonstrate that SIRA achieves a high attack success rate with minimal computational resources.

**Strengths:**

1.	The paper addresses a widely recognized problem in the field of text watermarking. By leveraging self-information, it enables effective watermark removal without compromising text quality.
2.	The experimental setup is comprehensive, covering various watermarking schemes and attack methods.

**Weaknesses:**

1.	Figures 3 and 4 are incorrectly placed, with Figure 4 missing some information, and table formats in this paper are inconsistent.
2.	The proposed method does not appear highly novelty, as it builds upon existing paraphrase attacks by using self-information to locate keywords.
3.	The experiments lack comparisons between self-information and other metrics, such as entropy and token probability, which could help establish the advantage of self-information.
4.	The proposed approach shares characteristics with watermark-stealing attacks [1,2,3], especially in the selection of keywords for targeted editing. A comparison with watermark-stealing attacks in both theoretical analysis and experiments would provide additional insights.
[1] N. Jovanović, R. Staab, and M. Vechev, “Watermark Stealing in Large Language Models.” http://arxiv.org/abs/2402.19361
[2] Q. Wu and V. Chandrasekaran, “Bypassing LLM Watermarks with Color-Aware Substitutions.” http://arxiv.org/abs/2403.14719
[3] Z. Zhang et al., “Large Language Model Watermark Stealing With Mixed Integer Programming.” http://arxiv.org/abs/2405.19677

**Questions:**

1.	What distinguishes self-information from entropy and probability, and what are the specific advantages of using self-information in this context?
2.	In Algorithm 1, is the model M in line 14 the attack model M_attack?
3.	In Figure 4(a), why does the word deletion method have a large impact on KGW-1’s PPL? Additionally, why is this impact much more significant than that of other methods in Figure 4(b)?
4.	In Table 1, the GPT Paraphraser shows a much higher attack success rate for Unigram watermarks than for DIPPER-based attacks, a phenomenon not observed with other watermarking methods. Additionally, SIR, a sentence embedding-based watermark scheme, should theoretically have robustness only second to Unigram, but this is not reflected in Table 1. Further discussion on these points is necessary.

---

> ### Author Response · Authors · 2024-11-19
> **Rebuttal part 1**
>
> We thank the reviewer 91eq for providing such comprehensive and constructive feedback. We address the concerns below:
>
> > **Q1**:  Figure placement and table format
>
> A1: We appreciate the reviewers' valuable examination and suggestions. We have fixed the issue in the  updated manuscript.
>
> > **Q2**: The difference between our work and other paraphrasing attacks
>
> A2:  We would like to take this opportunity to clarify that there are significant differences between our method and other paraphrasing attacks. Specifically, GPT Paraphraser simply instructs GPT model to paraphrase, while DIPPER utilizes a T5-XXL model fine-tuned specifically for paraphrasing. These paraphrasing attacks do not incorporate any special design; **current methods perform paraphrasing in a relative brute force manner**.
>
> The key insight of our method is that current watermarking techniques **require**  embedding in high-entropy/uniformly distributed tokens [1,2,3,4] to maintain text quality, as detailed in Section F. We are **the first to reveal that this inherent requirement can also serve as a potential vulnerability** and propose a method on how to exploit it in a black-box setting. Experiments validate the **effectiveness(SOTA) and efficiency** of our approach compared to all baseline methods . Our method can be applied to new models without requiring fine-tuning and works in challenging black-box settings. We believe our work offers valuable insights for developing more robust watermarking algorithms in the future. We would greatly appreciate it if the reviewer could provide any reference papers on methods leveraging similar ideas for paraphrasing attacks.
>
>
> > **Q3**:Difference with watermark-stealing attacks
>
> A3: We thank the reviewer for the valuable suggestion, we have added a new section H to discuss the difference in our revised manuscript and cited the mentioned papers.
>
> Watermark-stealing attacks [7,8,9] assume that attacker has:**unlimited access to the watermark generated model API** (including permission to modify hyperparameters), the detector API (with or with not under different assumption), knowledge of the context size and the watermarked model is aligned (capable of following instructions provided). These assumptions allow the attacker to execute multiple tries with design input prefix to probing the watermark algorithm.
>
> In a black-box paraphrasing attack, we assume that the attacker’s knowledge is limited to **only the watermarked text** and nothing else. This scenario is more challenging, and the assumptions are significantly weaker. *The frequency-based modification methods employed by watermark-stealing attacks approaches are entirely inapplicable in the black-box settings.
>
> We argue that  due to the **differing assumptions** underlying these two types of attacks, a fair comparison cannot be conducted. Recent studies on watermarks [1,2,3,4,5] do not incorporate such methods into robustness evaluations, and existing attack research [6] similarly do not conduct such comparisons due to the distinct problem setting.
>
> > **Q4** What distinguishes self-information from entropy and probability, and what are the specific advantages of using self-information in this context?
>
> A4:  We are thankful to the reviewer for this valuable question. Theoretical details and analysis please refer to appendix section F. In our preliminary experiments, we tested the direct use of both entropy and self-information for detection. Filtering using  entropy is also feasible, but self-information empirically outperforms entropy.
>
> We have conducted an extended experiment to filter based on self-information, entropy and probability. We follow the ablation setting using UPV as a watermark algorithm and set mask ratio to 0.7 for all three methods.  We use 50 samples of text and repeat 4 times to get the average attack success rate. The results are shown below
>
> | Method           | ASR  |
> | ---------------- | ---- |
> | Self-information | 94   |
> | Entropy          | 82   |
> | Probability      | 64   |
>
> The results show self-information is also empirically better than directly filtered by entropy and probability as we already mentioned in line 269.
>
> We infer the differences as self-information being a more sensitive, context-conditional metric, adapting to token sequences and scaling small probabilities linearly via log transformation. This context adaptability and sensitivity  make its empirical performance better.
>
> > **Q5** In Algorithm 1, is the model M in line 14 the attack model M_attack?
>
> A5: Yes,  we are thankful for the reviewer pointing it out. We have fixed this typo in our manuscript.

---

> > ### Author Response · Authors · 2024-11-19
> > **Rebuttal part 2**
> >
> > > **Q6**  In Figure 4(a), why does the word deletion method have a large impact on KGW-1’s PPL? Additionally, why is this impact much more significant than that of other methods in Figure 4(b)?
> >
> > A6: Upon verification, we found it is a transcription error in the word deletion data for KGW-1. In our initial draft, word deletion significantly increased the PPL of all watermarking methods, leading to overflow to NaN, which was not reflected in Figure 4a. We  have updated the revised Figure 4a 4b and Table 7 in the updated version.
> >
> > > **Q7**  In Table 1, the GPT Paraphraser shows a much higher attack success rate for Unigram watermarks than for DIPPER-based attacks, a phenomenon not observed with other watermarking methods. Additionally, SIR, a sentence embedding-based watermark scheme, should theoretically have robustness only second to Unigram, but this is not reflected in Table 1. Further discussion on these points is necessary.
> >
> > A7: We greatly appreciate the reviewer's suggestion. The use of GPT Paraphraser is simply providing an instruction for GPT to rewrite watermarked text. Since this process is entirely a black-box operation involves randomness, making it difficult to analyze why it works better in specific watermarks.  We can only ensure that our baseline experimental results are fair by conducting under the same setting with our method and employing a large sample size to ensure the reliability of the results.
> > Meanwhile theoretical robustness often differs from practical robustness across various attack methods, due to differences in experimental settings and randomness.We are concerned that  such discussions may lack generalizability among different set ups. For example, in [10], results indicate that UPV is more robust than SIR under two certain targeted attacks that align with our results, which also contradicts the theoretical robustness. We remain open to further discussion if the reviewer believes it is necessary.

---

> > > ### Author Response · Authors · 2024-11-19
> > > **rebuttal reference**
> > >
> > > [1] Kirchenbauer, J., Geiping, J., Wen, Y., Katz, J., Miers, I., and Goldstein, T. "A Watermark for Large Language Models." ICML, 2023.
> > >
> > > [2] Zhao, X., Ananth, P., Li, L., and Wang, Y.-X. "Provable Robust Watermarking for AI-Generated Text." ICLR, 2024.
> > >
> > > [3] Liu, A., Pan, L., Hu, X., Li, S., Wen, L., King, I., and Yu, P. "An Unforgeable Publicly Verifiable Watermark for Large Language Models." ICLR, 2023.
> > >
> > > [4] Lu, Y., Liu, A., Yu, D., Li, J., & King, I. . “An Entropy-based Text Watermarking Detection Method.” arXiv preprint arXiv:2403.13485, 2024
> > >
> > > [5] Hu, Z., Chen, L., Wu, X., Wu, Y., Zhang, H., and Huang, H. "A Semantic Invariant Robust Watermark for Large Language Models." arXiv preprint arXiv:2310.10669, 2023.
> > >
> > > [6] Krishna, K., Song, Y., Karpinska, M., Wieting, J., & Iyyer, M. “Paraphrasing evades detectors of ai-generated text, but retrieval is an effective defense.” NeruIPS , 2024
> > >
> > > [7] Jovanović, N., Staab, R., & Vechev, M.. “Watermark stealing in large language models.” arXiv preprint arXiv:2402.19361., 2024
> > >
> > > [8] Wu, Q., & Chandrasekaran, V. . “Bypassing LLM Watermarks with Color-Aware Substitutions”. arXiv preprint arXiv:2403.14719, 2024
> > >
> > > [9] Zhang, Z., Zhang, X., Zhang, Y., Zhang, L. Y., Chen, C., Hu, S., ... & Pan, S. “ Large Language Model Watermark Stealing With Mixed Integer Programming.” arXiv preprint arXiv:2405.19677., 2024
> > >
> > > [10] Pan, L., Liu, A., He, Z., Gao, Z., Zhao, X., Lu, Y., ... & Yu, P. S. . “Markllm: An open-source toolkit for llm watermarking.” arXiv preprint arXiv:2405.10051, 2024

---

> ### Author Response · Authors · 2024-11-22
> **Are the concerns of the Reviewer 91eq addressed?**
>
> Dear reviewer 91eq,
>
> We appreciate your feedback and concerns on our work. We strive to address your questions in our previous responses.  The key concerns were the self-information comparison with entropy and probability, the comparison with the watermark-stealing method, and the novelty of our proposed work. Our response above addresses those concerns. If there are specific parts that are unclear to the reviewer, we are more than happy to revise them.
>
> Please let us know if you have any remaining concerns.We thank you again for your valuable time and constructive feedback.
>
>
> Best,
> Authors

---

> ### Author Response · Authors · 2024-12-02
> **Did our response address the reviewer’s concerns?**
>
> Dear Reviewer 91eq,
>
> Thank you for your valuable feedback and the time you've devoted to reviewing our work. With the discussion period will end in 24 hours, we note that we have not yet received any further comments from you.
>
> If you feel your concerns have been adequately addressed, we kindly ask you to consider revising your score. If you have any remaining concerns, we would be glad to provide additional clarifications.
>
> Best regards, The Authors

---

### Official Review · Reviewer_EJ9Y · 2024-11-06

**Soundness:** 3
**Presentation:** 3
**Contribution:** 3
**Rating:** 6
**Confidence:** 4

**Summary:**

This paper introduce a novel watermark removal attack SIRA.
Current watermarking methods often favoring high-entropy tokens to embed watermark patterns. High-entropy tokens usually have high self-information. SIRA utilized self-information to identify potential ”green list” token candidates,  which are masked and then completed by LLM.

**Strengths:**

1 SIRA can be implemented using a more lightweight model compared to other model-based paraphrasing attacks.
2 This paper is well organized and discussions are relatively sufficient.

**Weaknesses:**

1 The semantic preservation of the proposed method is inferior compared to GPT Paraphraser.
A clerical error:Line 24 “tempering”? I think it should be "tampering".

**Questions:**

1 How to fairly evaluate the balance between the generated text quality and the attack effect?
2 Will the attacker really care so much about the resource reduction as shown in Table 2?

---

> ### Author Response · Authors · 2024-11-19
> **Rebuttal part 1**
>
> We are deeply thankful for reviewer EJ9Y valuable feedback. We clarify below the concerns of the reviewer:
>
> > **Q1**: The semantic preservation of the proposed method is inferior compared to GPT Paraphraser.
>
> A1: We agree with the reviewer’s opinion that  our method preserves semantics when using llama3-8b. However, we clarify that the attack method involves a trade-off between resource consumption, attack effectiveness, and semantic preservation. We would like to highlight that our attack performance significantly surpasses that of other baseline methods.
>
> In our experiment, we find that the performance of semantic preservation is highly influenced by the capabilities of the paraphrasing model. We use a lightweight Llam3-8b model to reduce resource requirements. As noted in Section E, replacing Llama3-8b with Llama3-70b in our setup increases the Semantic Preservation score by 16%.
>
> To further address the reviewer’s concerns, we conducted an additional experiment using GPT-4o as the paraphrasing model using our method while keeping all other settings unchanged. This adjustment resulted in a semantic preservation score of 8.02 which closely aligns with GPT Paraphraser(8.25).
>
> > **Q2**:  typo in line 124
>
> A2: We appreciate the reviewers' thorough examination. We have corrected  this typo in our revised manuscript.
>
>
> > **Q3**:  How to fairly evaluate the balance between the generated text quality and the attack effect?
>
> A3:  We are grateful if the reviewer further clarifies the question.We followed previous watermark works[2,3,4,5,6] using perplexity as the metric of text quality. **We clarify that text quality is not a problem for current paraphrasing attacks.** As we shown in Table 7, a notable conclusion is that compared to watermarked text without attack, **the text quality after being rewritten by our method and GPT Paraphraser is improved**. This conclusion has also been observed in recent studies [1]. Therefore we mostly focus on attack performance and efficiency together with semantic preservation rather than text quality.
>
> If the reviewer refers to the balance between semantic preservation and attack effectiveness. Evaluating this balance is an open question, depending on user objectives. We clarify that even the DIPPER method, which  with the lowest semantic preservation score , maintains sufficient semantic consistency in paraphrased text based on our human evaluation, this also reflected in its original paper human evaluation. Meanwhile as the data shown in Table 7, where s-bert represents the cosine similarity between the original and paraphrased texts. All Model-based methods have already achieved very high similarity scores which means the semantic is well-preserved. Hence, we believe the focus should shift to attack effectiveness. A more effective way is to employ more powerful LLM which increase semantic preservation and attack effects simultaneously.
>
>
> > **Q4**: Will the attacker really care so much about the resource reduction as shown in Table 2?
>
> A4: We appreciate the reviewer’s feedback. It is true that most attack methods focus solely on performance without considering resource reduction. However, we would like to emphasize that our method also demonstrates state-of-the-art attack performance. An effective method would be even better if it required less resources, wouldn't it?
>
> We would like to explain our motivation for reducing resource consumption from the perspectives of resource efficiency and scalability.
>
> 1. Lower the barrier to conduct watermarking research:  The current watermarking methods themselves require minimal resources, as a GPU capable of running a model like OPT-1.3b is sufficient for research.  However, verifying the robustness of the watermark demands significantly more resources, the mentioned DIPPER requires two A100 to operate.We believe that an effective and efficient method for testing watermark robustness which can run on consumer-level GPUs could lower the barrier to conduct watermarking research, thereby benefiting the community.
> 2. Scalability: As mentioned in Section E, we have included experiments using larger models such as Llama3-70b to demonstrate the scalability of our method. Using a larger model improves both the attack success rate and semantic preservation. If an attacker has sufficient resources, they could employ more powerful LLMs within our framework to enhance the results.

---

> > ### Author Response · Authors · 2024-11-19
> > **Rebuttal part 2**
> >
> > [1] He, Z., B. Zhou, H. Hao, A. Liu, X. Wang, Z. Tu, and Z. Zhang. "Can Watermarks Survive Translation? On the Cross-Lingual Consistency of Text Watermark for Large Language Models." ACL, 2024.
> >
> > [2] Zhao, X., P. Ananth, L. Li, and Y.-X. Wang. "Provable Robust Watermarking for AI-Generated Text." ICLR, 2024.
> >
> > [3] Pan, L., A. Liu, Z. He, Z. Gao, X. Zhao, Y. Lu, B. Zhou, and S. Liu. "MarkLLM: An Open-Source Toolkit for LLM Watermarking." EMNLP, 2024.
> >
> > [4] Kuditipudi, R., J. Thickstun, T. Hashimoto, and P. Liang. "Robust Distortion-Free Watermarks for Language Models." TMLR, 2023.
> >
> > [5] Kirchenbauer, J., J. Geiping, Y. Wen, J. Katz, I. Miers, and T. Goldstein. "A Watermark for Large Language Models." ICML, 2023.
> >
> > [6] Hu, Z., Chen, L., Wu, X., Wu, Y., Zhang, H., and Huang, H. "A Semantic Invariant Robust Watermark for Large Language Models." arXiv preprint arXiv:2310.10669, 2023.
> >
> > [7] Krishna, K., Song, Y., Karpinska, M., Wieting, J., & Iyyer, M. “Paraphrasing evades detectors of ai-generated text, but retrieval is an effective defense.” NeruIPS , 2024

---

> ### Author Response · Authors · 2024-11-22
> **Are the concerns of the Reviewer EJ9Y addressed?**
>
> Dear reviewer EJ9Y,
>
> We appreciate your feedback and concerns. We strive to address your questions in our previous responses. The key concerns were the text quality and motivation of resource reduction. If there are specific parts that are unclear to the reviewer, we are more than happy to revise them.
>
> Please let us know if you have any remaining concerns. We thank you again for your valuable time and constructive feedback.
> Best, Authors

---

> ### Author Response · Authors · 2024-12-02
> **Did our response address the reviewer’s concerns?**
>
> Dear Reviewer EJ9Y,
>
> Thank you for your valuable feedback and the time you've devoted to reviewing our work. With the discussion period will end in 24 hours, we note that we have not yet received any further comments from you.
>
> If you feel your concerns have been adequately addressed, we kindly ask you to consider revising your score. If you have any remaining concerns, we would be glad to provide additional clarifications.
>
> Best regards,
> The Authors

---

> > ### Comment · Reviewer_EJ9Y · 2024-12-03
> > **Thanks**
> >
> > Thank you for your response. It partly addresses my concerns.  Therefore, I will raise my rating.

---

> > > ### Author Response · Authors · 2024-12-03
> > >
> > > Many thanks for your support, and thank you again for reviewing our paper!

---

### Author Response · Authors · 2024-11-19
**common reply**

Dear AC and reviewers,

We sincerely thank all reviewers and the Area Chair for their valuable time and insightful feedback. We are pleased that Reviewers 91eq and ATEL have recognized **the effectiveness of our method**, and that **the low resource requiremen** of our approach has been acknowledged by EJ9Y and ATEL. Additionally, EJ9Y and 91eq found our **experiments and discussions to be comprehensive**, and Reviewer AqqW acknowledged the advantage of **our method being effective in a black-box setting**.

We address some common concerns raised by the reviewers below.

> **Q1**: Novelty and difference with other paraphrasing attacks

A1: Current paraphrasing simply instructs the model to perform paraphrasing in a relative brute force manner. The key insight of our method is that watermarking algorithm **require** embedding in high-entropy tokens [1,2,3,4] to maintain text quality, as detailed in Section F. We are the first to  **reveal that this necessary requirement can also serve as a potential vulnerability** and propose an effective way on **how to exploit it in a black-box setting**. Our method achieves the best empirical attack performance while requiring minimal resources which makes it a suitable method to evaluate watermark robustness . We believe our work provides insights for developing more robust watermarking algorithms in the future and serves as an easy to use robustness evaluation tool to benefit the community.

> **Q2**: Shorter execution time

A2: The total time consumption for SIRA consists of two parts: two generations by the base model and  the self-information mask. The self-information mask is nearly negligible, as it does not require any text generation (less than 0.1 seconds).  The other two generations take around 5 seconds per generation on a single A100 GPU. Thus the total execution time is around 10 seconds. We use huggingface library in our experiment.

The DIPPER method utilizes a specially fine-tuned T5-XXL model for text paraphrasing. This model needs at least 40 GB VRAM to run and one time generation requires around 15 seconds on two A100GPU. DIPPER relies on this specific fine-tuned model, preventing it from transferring to a smaller model.  We use DIPPER official open-source code and weights  in our experiment.

For GPT,  We use gpt-4o-2024-05-13 in our experiment.  In response to Reviewer AqqW's request, we conducted retests at different times of the day and reported the average results.The execution time is 12.6±0.4 seconds

All above mentioned we set max output token number is 256.

> **Q3**: Can we achieve better semantic preservation?

A3: The answer is yes. As the fill in blank prompt we showed in section D, a more powerful paraphrasing will better understand the prompts and its better capability will lead to higher semantic preservation. Our work achieves state-of-art performance in black box paraphrasing attacks and less resource requirement. We clarify that the attack method involves a trade-off between resource consumption, attack effectiveness, and semantic preservation. For example, the white-box watermark attack method RandomWalk[5] theoretically removes any watermark completely. However, it does not guarantee preserving any semantics and requires multiple models; each watermarked text takes approximately 20 minutes to attack on 3 A100 GPU. Our motivation is to propose an easy-to-use, low-resource, and effective tool to further advance robust watermark research, thus we choose to present our work on the lightweight model Llama3-8b.

In response to all feedback received, we have updated our manuscript further, marking all changes in $\color{red}red$\. All updates  and discussion will be included in the revised manuscript.  We sincerely thank you for your suggestions to improve our manuscript.

---

> ### Author Response · Authors · 2024-11-19
> **common reply reference**
>
> [1] Kirchenbauer, J., J. Geiping, Y. Wen, J. Katz, I. Miers, and T. Goldstein. "A Watermark for Large Language Models." ICML, 2023.
>
> [2] Zhao, X., P. Ananth, L. Li, and Y.-X. Wang. "Provable Robust Watermarking for AI-Generated Text." ICLR, 2024.
>
> [3] Liu, A., L. Pan, X. Hu, S. Li, L. Wen, I. King, and P. Yu. "An Unforgeable Publicly Verifiable Watermark for Large Language Models." ICLR, 2023.
>
> [4] Lu, Y., A. Liu, D. Yu, J. Li, and I. King. "An Entropy-Based Text Watermarking Detection Method." ACL, 2024.
>
> [5] Zhang, H., B. Edelman, D. Francati, D. Venturi, G. Ateniese, and B. Barak. "Watermarks in the Sand: Impossibility of Strong Watermarking for Generative Models." ICML, 2024.

---

### Author Response · Authors · 2024-11-25
**Are the concerns of the reviewer addressed?**

Dear Reviewers,

Thank you for your comprehensive and constructive reviews. Your feedback is invaluable for improving our manuscript.

 Given that the discussion window is closing soon, please let us know if you have any remaining concerns, *we are really looking forward to hearing from you further*. If you are satisfied with our responses and revisions,, we would appreciate it if you increase your score to reflect our revised manuscript.

Thank you once again for your time and help. Wishing you a wonderful day!

Best Regards,

The Authors

---

### Author Response · Authors · 2024-11-29
**We are looking forward to your further feedback**

Dear Reviewers,

We sincerely appreciate your valuable feedback on our submission. As the extended discussion phase is about to close, we kindly wish to follow up, as we have not yet received any further comments or acknowledgment regarding our latest updates and responses.

Thank you for your time and consideration, and we look forward to your response.

Best regards,
Authors

---

### Author Response · Authors · 2024-12-04
**Summary of Reviews and Responses**

**Summary of our paper**:Our paper introduces the Self-Information Rewrite Attack , a method that exploits vulnerabilitiy in current text watermarking algorithms that require embed pattern in high-entropy tokens . By identifying high-entropy tokens where watermarks are typically embedded, SIRA effectively removes these watermarks through a **black-box** approach.

**Experimental Results**:Experimental results demonstrate that SIRA achieves **SOTA** attack performance with over 90% attack success rate in 6 recent watermark removal while maintaining efficiency in execution time and computational resources. Our method achieves a **30%-60% improvement** in attack success rates and **50% reduction in cost** compared to previous algorithms,

**Contribution**: Our paper suggesting **critical vulnerabilities** in current watermarking techniques. Meanwhile our algorithm **change the brute-force paradigm of exisiting paraphrasing attacks**, reducing resource consumption and effectively lowering the requirements for watermark research. It has the potential to become an easy-to-use tool for testing watermark robustness.


-------
We appreciate that reviewers found our proposed method **effective(91eq,ATEL), efficient(EJ9Y,ATEL)**  and considered the paper well-written,experiments are **comprehensive(EJ9Y,91eq)** . Our paper **addresses a widely recognized problem** in the field of text watermarking(91eq).

**Key initial concerns included**:
> **Semantic preservation(EJ9Y,Aqqw)**

We clarify that semantic preservation is related to the performance of the rewriting model. For efficiency, we opted for a small model in our initial paper. To resolve the concern by the reviewers, we included experiments with Llama3-70B and ChatGPT-4, demonstrating that stronger models significantly enhance semantic preservation.

> **Motivation to reduce resources requirement (EJ9Y)**

We clarify that our goal is to reveal vulnerabilities in watermarking methods and lower research barriers, while demonstrating our method's scalability through experiments.

> **Novelty compared to current Paraphrasing Attack(91eq)**

We clarify that current paraphrase attacks are untargeted and brute-force, while our method is more controllable, effective, and achieves SOTA performance according to our experiment.

> **Comparison with watermark stealing attack(91eq)**

We added relevant sections to the appendix as per the reviewer's suggestions,we emphasize that our method is fundamentally different from watermark stealing, as evidenced by the fact that such methods cannot operate in black-box settings.
> **The advantage of self-information compared to probability and entropy**(91eq)

We added the requested ablation experiments and included theoretical explanations in the corresponding appendix section.
> **Ablation regarding iterative paraphrasing(AqqW)**

We have added the requested ablation experiments as per the reviewer’s suggestions.The experimental results demonstrate the effectiveness of our method.
> **How can SIRA achieve shorter time and comparison with GPT Paraphraser(AqqW,ATEL)**


Following the reviewer’s suggestions, we conducted a more rigorous experiment and included a cost analysis for using third-party services.

> **Results are inconsistency(ATEL)**

We emphasize that the inconsistency mentioned by the reviewer refers to two different experiments, and thus, this concern is **not valid**.

-------
**During the rebuttal period**

**Reviewer EJ9Y** acknowledged our contributions, increasing the contribution score from 2 to 3 and raising the overall rating from 5 to 6.

**Reviewer AqqW** decided to keep the score at 5.

**Reviewer 91eq, ATEL** has not responded to our rebuttal.

-------

We have added the experiments requested by the reviewers and provided further clarification on our novelty and any potential misunderstandings. Unfortunately, during the rebuttal period, we did not receive additional feedback from Reviewer ATEL or Reviewer 91eq. However, we are confident that we have addressed all the concerns raised by reviewers. **We would like to sincerely thank all the reviewers and the AC for their valuable comments and feedback, which have helped us improve the quality of our paper**.

---

### Note · Authors · 2025-01-23

I have read and agree with the venue's withdrawal policy on behalf of myself and my co-authors.